# Action Manifold Smoothing: A Lipschitz Pathway Perspective on High-Dimensional Reinforcement Learning

Zhihao Lin [1]

## Abstract

High-dimensional continuous control remains challenging in deep reinforcement learning, where algorithms like TD3 and SAC often collapse. We propose a unifying **Lipschitz Pathway** framework that decomposes instability into four amplification stages, namely action parameterization ($L_1$), dynamics sensitivity ($L_2$), Q-network curvature ($L_3$), and temporal-difference (TD) target stability ($L_4$), where errors compound multiplicatively along the learning pipeline. Our analysis identifies a *discrete-continuous mismatch* as the root cause: value functions trained from sparse point samples must generalize over continuous manifolds, leading to multiplicative error amplification along the pathway. To address this, we introduce **Action Manifold Smoothing (AMS)**, which replaces point-wise TD targets with orthogonally-sampled neighborhood averages, jointly regularizing $L_3$ (via implicit Laplacian smoothing) and $L_4$ (via local manifold supervision). We further characterize when Lipschitz-constrained Q-networks and geometric action priors are beneficial based on task structure. Empirically, AMS enables both TD3 and SAC to achieve over 400 reward on the 38-D Dog Run task within 1M steps, where baselines fail. These results validate the Lipschitz pathway as a principled framework for diagnosing and solving stability bottlenecks in high-dimensional control.

## 1. Introduction

Deep reinforcement learning (RL) has achieved remarkable success (OpenAI et al., 2019) in continuous control, yet a fundamental tension remains: **RL algorithms learn contin-**

**uous functions from discrete samples**. At each timestep, the agent observes a single state-action pair, corresponding to a measure-zero in continuous space. The Q-function must generalize across the entire manifold, filling gaps through extrapolation. When this fails, learning becomes unstable: Q-values diverge, gradients chase artifacts, and training collapses (Sutton & Barto, 1998). This *discrete-continuous mismatch* is the root cause of instability in model-free RL, particularly in high-dimensional action spaces.

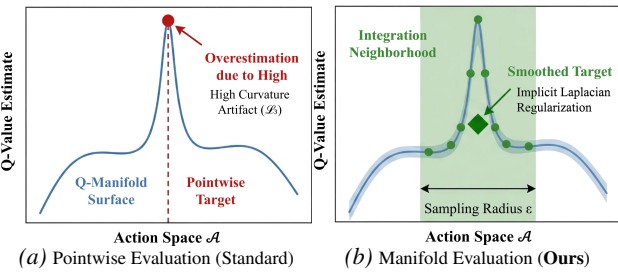

*Figure 1.* Geometric intuition of Action Manifold Smoothing (AMS). (a) Standard pointwise evaluation is sensitive to sharp, spurious peaks (high-curvature artifacts, $L_3$) in the Q-landscape. (b) AMS suppresses such artifacts via local neighborhood averaging, yielding smoothed targets that are robust to local irregularities.

The field has accumulated numerous stabilization techniques, including target networks, clipped double Q-learning, policy smoothing, and ensembles (Chen et al., 2021). Empirical studies further suggest that code-level optimizations and implementation details can dominate performance in deep RL (Engstrom et al., 2020). However, these mechanisms are largely heuristic and lack a unified geometric grounding. For instance, the effectiveness of Gaussian target noise (Fujimoto et al., 2018), the widespread use of fixed clipping thresholds (e.g., 0.5) (Fujita & Maeda, 2018), and the near-universal adoption of tanh squashing in policy outputs (Haarnoja et al., 2018) are justified empirically rather than from first principles. As a result, such solutions often exhibit brittle, environment-dependent behavior.

Geometric perspectives on representation and value approximation have long been explored in reinforcement learning (Bellemare et al., 2019). However, how geometric irregularities propagate through the action-value learning pipeline remains less understood. We propose a unifying perspec-

[1]James Watt School of Engineering, University of Glasgow, Glasgow, UK. Correspondence to: Zhihao Lin <kk43327897@gmail.com>.

*Proceedings of the $43^{rd}$ International Conference on Machine Learning*, Seoul, South Korea. PMLR 306, 2026. Copyright 2026 by the author(s).

tive: **instability arises when small perturbations amplify through the learning pipeline**. This can be formalized as a Lipschitz pathway:

$$\pi \xrightarrow{L_1} a \xrightarrow{L_2} s' \xrightarrow{L_3} Q(s', a') \xrightarrow{L_4} y \qquad (1)$$

where each $L_i$ denotes the Lipschitz constant at stage $i$: $L_1$ captures action parameterization geometry (e.g., $\tanh$ squashing), $L_2$ environment dynamics sensitivity, $L_3$ Q-network curvature, and $L_4$ TD target computation stability. Stable learning requires controlling the total amplification $L_1 \times L_2 \times L_3 \times L_4$. This clarifies why different environments require different techniques: each method constrains a different $L_i$, and tasks differ in which factors dominate.

Building on this insight, we propose **Action Manifold Smoothing (AMS)** (Figure 1), a framework that applies geometric constraints to the critical downstream stages $L_{3-4}$. Specifically, AMS replaces pointwise TD evaluation with *geometry-aware neighborhood averaging* to regularize Q-curvature, enforces implicit Lipschitz bounds via *network normalization*, and aligns *action projection geometry* ($\tanh$ vs. normalize) with the task's physical structure.

Our framework reveals a duality with model-based RL: Dreamer smooths state transitions ($L_2$), while AMS smooths the Q-landscape ($L_{3-4}$). Both address the discrete–continuous mismatch from opposite ends of the pathway. AMS is algorithm-agnostic, improving both Twin Delayed Deep Deterministic policy gradient (TD3) (Fujimoto et al., 2018) and Soft Actor-Critic (SAC) (Haarnoja et al., 2018). Notably, SAC's entropy regularization constrains $L_1$ (policy diversity) but not $L_3$ (value smoothness), explaining why entropy alone is insufficient to stabilize learning in high-dimensional action spaces. By disentangling exploration from geometric stability, AMS enables both algorithms to succeed where they previously failed.

We validate on challenging locomotion tasks. On high-dimensional control problems, such as the 38-dimensional Dog Run task (Tassa et al., 2018), which is known to exhibit extreme variance and instability in model-free learning regimes (Agarwal et al., 2021), AMS achieves over 400 reward within 1M environment steps, where vanilla TD3 and SAC fail to learn meaningful behaviors. Ablations confirm the framework's predictions: high-dimensional tasks require the full $L_1 + L_3 + L_4$ stack. We summarize our main contributions as follows:

- **Diagnostic Framework:** We identify the discrete-continuous mismatch as a primary source of instability and propose the Lipschitz signal pathway ($L_1$–$L_4$) to unify and explain existing stabilization techniques.

- **Theoretical Insight:** We prove that Neighborhood TD (NTD) imposes local Lipschitz constraints through an

*Table 1.* Stabilization techniques under the **Lipschitz pathway**.

| Method | Target | Mechanism | Stage |
|---|---|---|---|
| *Discretization (Geometry Avoidance)* | | | |
| Action Quantization | Action space | Binning to classification | – |
| *Action Space Geometry ($L_1$)* | | | |
| GAC | Action projection | Spherical constraint | $L_1$ |
| SAC Entropy | Policy | Entropy regularization | $L_1$ |
| TD3 Smoothing | Target action | Noise injection | $L_1$ |
| Diffusion Policy | Distribution | Iterative denoising | $L_1$ |
| *Dynamics Smoothing ($L_2$)* | | | |
| DrQ | State repr. | Augmentation | $L_2$ |
| Dreamer | Dynamics | World model | $L_2$ |
| TD-MPC2 | Dyn + Value | Model + planning | $L_{1\text{-}3}$ |
| *Q-Network Regularization ($L_3$)* | | | |
| LayerNorm | Activations | Normalization | $L_3$ |
| Spectral Norm | Weights | Lipschitz bound | $L_3$ |
| REDQ | Ensemble | Model averaging | $L_3$ |
| C51 / Distributional | Q representation | Return distribution | $L_3$ |
| *TD Target Smoothing ($L_4$)* | | | |
| Target Network | Weights | Polyak averaging | $L_4$ |
| Clipped Double Q | Q-estimate | $\min(Q_1, Q_2)$ | $L_4$ |
| Delayed Policy Update | Update frequency | Reduce actor updates | $L_4$ |
| **AMS (Ours)** | **Q-landscape** | **Neighborhood avg.** | **$L_{3,4}$** |

implicit Laplacian effect, providing variance reduction and curvature regularization guarantees.

- **Empirical Breakthrough:** We demonstrate that AMS enables both deterministic (TD3) and stochastic (SAC) algorithms to solve high-dimensional tasks (e.g., Dog Run, 38D) where baselines fail, highlighting geometric smoothing as a key factor for scalable control.

**Conflict of Interest Disclosure.** The authors declare no financial conflicts of interest. This work was conducted in an academic setting and does not evaluate any commercial product developed by an entity employing the authors.

## 2. Related Work

We organize prior stabilization techniques through our Lipschitz pathway (Eq. 1), summarized in Table 1.

**Discretization.** Action quantization sidesteps continuous geometry by discretizing actions into atomic bins, effectively converting the problem into classification (Tang & Agrawal, 2020; Zhu et al., 2025). While this avoids action-space extrapolation, it sacrifices fine-grained control and scales poorly to high-dimensional settings, representing geometric evasion rather than true resolution.

**Action Space Geometry ($L_1$).** Geometric structure in reinforcement learning has traditionally been explored in the policy parameter space, e.g., via Natural Policy Gradient methods (Müller & Montúfar, 2024). *In contrast*, our framework focuses on the explicit geometry of the action output space ($L_1$). SAC (Haarnoja et al., 2018) regularizes policy entropy, encouraging action diversity but imposing no

constraint on Q-geometry. This explains its failure on high-dimensional tasks despite entropy regularization. TD3's target policy smoothing (Fujimoto et al., 2018) adds noise to the action before Q-evaluation; while often categorized as target smoothing, this operates at $L_1$ (perturbing the query point) rather than $L_4$ (smoothing the Q-landscape). Geometric Action Control (GAC) (Lin, 2026) constrains policy outputs to unit spheres, while diffusion policies (Chi et al., 2024) shape action distributions through iterative denoising, imposing explicit geometric structure on the policy output. By comparison, AMS targets geometric irregularities that emerge implicitly in the value function landscape ($L_{3-4}$).

**Dynamics Smoothing ($L_2$).** Model-based methods address instability by learning smooth dynamics, often inducing broader smoothing effects across multiple stages of the Lipschitz pathway. Dreamer (Hafner et al., 2025) learns a latent world model for imagined rollouts; TD-MPC2 (Hansen et al., 2024) combines model learning with planning, jointly smoothing $L_{1-3}$. DrQ (Yarats et al., 2021) uses state augmentation to regularize representations, achieving representation-level smoothing without explicit modeling. These approaches reduce upstream sensitivity ($L_2$), complementing downstream value smoothing ($L_{3-4}$).

**Q-Network Regularization ($L_3$).** Spectral normalization (Miyato et al., 2018) explicitly bounds weight matrix norms to control $\text{Lip}(Q)$, a principle broadly advocated in Lipschitz-aware RL (Gogianu et al., 2021). LayerNorm (Ba et al., 2016) constrains activations, providing implicit Lipschitz regularization. REDQ (Chen et al., 2021) trains ensembles of Q-networks, reducing variance through model averaging. Distributional RL methods like C51 (Bellemare et al., 2017) learn full return distributions, enriching value representation but still performing pointwise Bellman backups. Recent theoretical work (Tiwari et al., 2025) proves that neural policies induce low-dimensional state manifolds whose dimensionality is bounded by that of the action space. This geometric perspective suggests that value functions are effectively defined over structured, low-dimensional subsets of the state–action space, providing theoretical support for the design of manifold-aware regularization.

**TD Target Smoothing ($L_4$).** Target networks (Mnih et al., 2015) stabilize learning through averaging of weights. Clipped double Q-learning (Fujimoto et al., 2018) takes $\min(Q_1, Q_2)$ to reduce overestimation. Delayed policy updates in TD3 reduce actor optimization frequency, stabilizing the TD target ($L_4$). These methods constrain $L_4$ by stabilizing the TD target computation. AMS controls $L_4$ via neighborhood averaging with implicit curvature regularization (Theorem 3.3), providing geometric smoothing of the value landscape rather than temporal stabilization alone.

**Positioning.** Existing methods typically constrain a single $L_i$ factor. Our framework reveals their complementary roles: environments differ in which factors are large, explaining task-dependent effectiveness. AMS targets downstream stages $L_{3-4}$, where errors corrupt gradient updates, providing principled stabilization for high-dimensional RL.

## 3. Method

### 3.1. Preliminaries

Consider an RL agent in continuous spaces $\mathcal{S} \subseteq \mathbb{R}^{d_s}$ and $\mathcal{A} \subseteq \mathbb{R}^{d_a}$ governed by dynamics $p(s'|s, a)$ and rewards $r(s, a)$. Actor-critic methods maximize the discounted return ($\gamma \in [0, 1)$) by learning a policy $\pi_\phi$ and a parametric Q-function $Q_\theta$ via TD learning. Using target networks $\{Q_{\theta'}, \pi_{\phi'}\}$, the TD target is:

$$y = r + \gamma Q_{\theta'}(s', \pi_{\phi'}(s')). \tag{2}$$

Our analysis applies to TD3, SAC, and related algorithms.

### 3.2. The Discrete-Continuous Mismatch

The fundamental challenge of value-based deep RL lies in a representational mismatch: **training data arrive as discrete point samples, yet the Q-function must generalize as a continuous surface**. At each timestep, the agent observes a single $(s, a, r, s')$ tuple, a measure-zero event in the continuous state-action space. The Q-network must extrapolate from these isolated points to the entire manifold. This extrapolation becomes increasingly unreliable in high dimensions. In $d_a$-dimensional action space, the expected nearest-neighbor distance among $N$ samples scales as $O(N^{-1/d_a})$: coverage degrades exponentially with dimension, leaving vast regions where the Q-function is unconstrained by data. Learning at one action provides virtually no information about Q-values at nearby unvisited actions.

The consequence is geometric instability. Consider the TD target $y = r + \gamma Q(s', \pi(s'))$. If the Q-function exhibits high curvature between training points, the target may land on a spurious extremum, namely a region where Q-values vary rapidly under small action perturbations. Policy gradients then chase these artifacts, destabilizing learning.

From our Lipschitz pathway perspective (Eq. 1), this instability manifests as uncontrolled growth of $L_3$, the Lipschitz constant $\text{Lip}(Q(s, \cdot))$, in sparsely supervised regions. Small perturbations in action induce large variations in the TD target, which are further amplified through gradient backpropagation. The core issue is not insufficient exploration, but the absence of geometric constraints on the value landscape. *What we need is not a Q-function that memorizes training points, but one that remains smooth over neighborhoods the policy may visit*, so that TD targets reflect robust value estimates rather than extrapolation artifacts.

## 3.3. Sensitivity Analysis of TD Learning

We analyze TD learning as a propagation process. A perturbation $\delta a$ in action propagates through the learning pipeline:

$$\delta a \xrightarrow{L_{\text{dyn}}} \delta s' \xrightarrow{L_Q} \delta y \longrightarrow \delta \nabla_\theta, \quad (3)$$

where $L_{\text{dyn}}$ is the environment's sensitivity to actions, and $L_Q$ the Q-network's Lipschitz constant. When these factors are large, small action perturbations induce large TD errors.

**Proposition 3.1** (Sensitivity Amplification). *Let $L_{\text{dyn}}$ denote the Lipschitz constant of the environment dynamics $p(s'|s, a)$ with respect to $a$, and $L_Q$ the Lipschitz constant of the Q-network. A perturbation $\|\delta a\| = \epsilon$ induces TD target error bounded by $O(L_{\text{dyn}} \cdot L_Q \cdot \epsilon)$.*

This result (proof in Appendix A.1) explains why different tasks require different stabilization strategies. Environments with sensitive dynamics (e.g., balancing) exhibit large $L_{dyn}$ and therefore amplify errors along the TD pathway, benefiting from explicit constraints on Q-smoothness. In contrast, tasks with naturally smooth dynamics may only require action-space regularization. More generally, this proposition formalizes our Lipschitz pathway perspective: total sensitivity scales multiplicatively along the chain, and stability can be achieved by constraining *any* factor. AMS targets the downstream factors $L_3$ and $L_4$, which we find particularly effective, as errors at these stages directly corrupt gradient updates before any opportunity for downstream correction.

## 3.4. Action Manifold Smoothing

We now present **AMS**, a framework for controlling sensitivity at the action–value interface by replacing pointwise TD targets with local manifold expectations.

**From Pointwise TD to Manifold Supervision.** The standard TD target evaluates the value function at a single action:

$$y_{\text{td}} = r + \gamma Q(s', \mu'), \ \mu' \triangleq \pi(s') \text{ (target policy)}. \quad (4)$$

As established in Section 3.2, this point estimate is vulnerable to local irregularities of the Q-surface, precisely the $L_4$ instability in our Lipschitz pathway.

We propose replacing pointwise evaluation with a **neighborhood expectation** centered at the policy output:

$$y_{\text{ntd}} = r + \gamma \int_{\mathcal{A}} Q(s', a) \, p_{\text{geo}}(a \mid \mu') \, da, \quad (5)$$

where $p_{\text{geo}}(\cdot \mid \mu')$ is a geometry-aware distribution concentrated near the policy output $\mu' = \pi(s')$. This formulation admits a continuous interpretation as a local convolution over the action manifold (Appendix A.2), shifting TD learning from *point supervision* to *manifold supervision*: the Bellman backup integrates over a local action region.

In practice, we approximate this integral via Monte Carlo sampling in an $\epsilon$-neighborhood, using *orthogonal directions* to efficiently cover the local action manifold:

$$y_{\text{ntd}} \approx r + \gamma \cdot \frac{1}{K} \sum_{k=1}^{K} Q(s', a'_k), \quad a'_k \sim p_{\text{geo}}(\cdot \mid \mu'). \quad (6)$$

*Geometric intuition.* Imagine the Q-function as a rugged landscape and the policy as a surveyor standing on it. A point estimate asks: "what is the height exactly under your feet?" If the surveyor stands on a narrow spike or pit, this measurement becomes unreliable. NTD instead asks for the average height in the vicinity, analogous to estimating local elevation from nearby measurements. Local irregularities are suppressed, while the dominant value trend is preserved.

*Theoretical Guarantees.* NTD provides both variance reduction and implicit curvature regularization.

**Theorem 3.2** (Variance Reduction). *Let $Q$ be $L$-Lipschitz in action. For $K$ i.i.d. samples $\{a'_k\}_{k=1}^K$ drawn from an $\epsilon$-neighborhood around $\mu' = \pi(s')$, the variance of the NTD target (due to sampling randomness) satisfies:*

$$\text{Var}[y_{\text{ntd}}] \leq \frac{\gamma^2 L^2 \epsilon^2}{K}. \quad (7)$$

*As $K \to \infty$, the neighborhood estimate converges to the local expectation with vanishing variance, while the implicit Laplacian correction (Theorem 3.3) provides geometric smoothing of the Q-landscape.*

**Proof Sketch.** *Lipschitz continuity bounds the Q-value range within an $\epsilon$-ball by $2L\epsilon$. Averaging $K$ samples reduces variance by a factor of $K$, while the additive term captures the bias introduced by neighborhood size. See Appendix A.3 for the full proof.*

**Theorem 3.3** (Implicit Hessian Regularization). *Let $Q(s, \cdot)$ be twice continuously differentiable. Under sampling over $K$ mutually orthonormal directions on the sphere $\|\mathbf{v}\| = \epsilon$ centered at $\mu'$, the expected target approximates:*

$$\mathbb{E}[y_{\text{ntd}}] \approx r + \gamma(Q(s', \mu') + \frac{\epsilon^2}{2d_a} \Delta_a Q(s', \mu') + O(\epsilon^4)), \quad (8)$$

*where $\Delta_a Q = \text{tr}(\nabla_a^2 Q)$ is the Laplacian of $Q$ with respect to action.*

*Interpretation.* The Laplacian $\Delta_a Q$ measures local curvature: $\Delta_a Q > 0$ at local minima (valleys) and $\Delta_a Q < 0$ at local maxima (peaks). The correction term acts as **geometric diffusion**, subtracting value from peaks and adding to valleys, which is mathematically equivalent to evolving $Q$ under the heat equation (Bishop, 1995) for time $O(\epsilon^2)$. In terms of our Lipschitz pathway, this provides implicit regularization of $L_3$ (Q-network smoothness) through the choice of TD target ($L_4$), yielding a single mechanism that

jointly controls two stages, akin to damping. Proof in Appendix A.4.

*Remark* 3.4 (Policy Entropy Does Not Imply Value Smoothness). SAC regularizes policy entropy via $J_{\text{SAC}} = \mathbb{E}[Q(s,a)] + \alpha \mathcal{H}(\pi)$, which acts solely on the policy distribution, effectively constraining $L_1$. However, stability along the TD pathway depends on controlling the *entire* sensitivity chain $L_1$–$L_4$. Regularizing $L_1$ encourages diverse action sampling but imposes no constraint on the geometry of the Q-function ($L_3$). Even under a high-entropy policy, $Q(s,\cdot)$ may exhibit sharp peaks or large curvature. Consequently, small action perturbations can still induce large TD errors. In contrast, NTD explicitly regularizes the Q-landscape via the implicit Laplacian term $\frac{\epsilon^2}{2d_a}\Delta_a Q$, directly constraining $L_3$. This reveals a structural limitation of entropy-based methods: **constraining $L_1$ (policy diversity) does not imply constraining $L_3$ (value smoothness)**.

**Orthogonal Sampling.** Standard isotropic Gaussian noise often wastes samples on redundant directions. In high-dimensional spaces, random vectors tend to align rather than spread uniformly (Vershynin, 2018).

We propose **orthogonal sampling**: given the anchor action $\mu' = \pi(s')$, we construct $K$ perturbations via QR decomposition of a random Gaussian matrix to obtain orthonormal directions:

$$a'_k = \mu' + \epsilon \mathbf{v}_k, \quad \text{s.t. } \mathbf{v}_i^\top \mathbf{v}_j = \delta_{ij}. \tag{9}$$

Geometrically, these directions span a $K$-dimensional subspace of the local tangent space at $\mu'$. By enforcing orthonormality, AMS ensures maximal coverage of the neighborhood with minimal redundancy, yielding an efficient approximation of the manifold integral. This observation is consistent with results from structured Monte Carlo methods (Choromanski et al., 2019), which show that orthogonality can improve estimator quality in high-dimensional kernel approximation. Formally, orthogonal unit vectors maximize the volume of the spanned parallelotope and eliminate cross-covariance in Q-function estimation, properties absent in random sampling (see Appendix A.5).

**Lipschitz-Constrained Q-Networks.** NTD controls sensitivity at the TD target level ($L_4$). However, if the Q-network has steep gradients, it amplifies remaining perturbations, corresponding to $L_3$ in Eq. (1). We therefore regularize the Q-network's Lipschitz constant directly.

*LayerNorm as implicit $L_3$ constraint.* Explicit Lipschitz constraints such as Spectral Normalization (Gogianu et al., 2021) have been proposed for stabilizing value networks. Alternatively, we find that adding LayerNorm to Q-networks (**LN-Q**) (Wang et al., 2025) provides sufficient implicit regularization in practice, improving stability

---

**Algorithm: Action Manifold Smoothing for TD3**

**Hyperparameters:** smoothing radius $\epsilon$; neighbors $K$; discount $\gamma$; policy delay $d_{\text{policy}}$; soft update rate $\tau$

**Input:** replay buffer $\mathcal{B}$

**Output:** actor $\pi_\phi$ and critics $\{Q_{\theta_j}\}_{j=1}^2$

1. Sample minibatch $(s, a, r, s', \texttt{done})$ from $\mathcal{B}$
2. Compute anchor action: $\mu' \leftarrow \pi_{\phi'}(s')$
   *(deterministic target action)*
3. Sample $K$ Gaussian vectors and orthogonalize via QR decomposition
   $\{\mathbf{v}_k\}_{k=1}^K \leftarrow \texttt{QR}(\mathcal{N}(0, I))$
4. Construct neighborhood actions:
   $a'_k \leftarrow \texttt{clip}(\mu' + \epsilon \cdot \mathbf{v}_k)$
5. Compute NTD target:
   $y \leftarrow r + \gamma(1 - \texttt{done}) \cdot \frac{1}{K} \sum_{k=1}^K \min_{j=1,2} Q_{\theta'_j}(s', a'_k)$
6. Update critics by minimizing Bellman error:
   $\theta_j \leftarrow \arg\min_{\theta_j} \mathbb{E}[(Q_{\theta_j}(s, a) - y)^2]$
7. **Every $d_{\text{policy}}$ steps:** update actor
   $\phi \leftarrow \arg\max_\phi \mathbb{E}[Q_{\theta_1}(s, \pi_\phi(s))]$
8. Soft-update target networks:
   $\theta'_j \leftarrow \tau\theta_j + (1-\tau)\theta'_j, \quad \phi' \leftarrow \tau\phi + (1-\tau)\phi'$

*Figure 2.* AMS-TD3 algorithm with action manifold smoothing.

---

in high-dimensional tasks while integrating naturally with NTD. LayerNorm bounds intermediate activations, limiting $\text{Lip}(Q)$ with respect to input perturbations (Appendix A.6).

*When to use LayerNorm.* LayerNorm trades resolution for stability. It is most beneficial when dynamics sensitivity $L_{\text{dyn}}$ remains high across dimensions, as in tightly coupled locomotion. In tasks with decoupled actions or localized sensitivity, NTD alone often suffices.

*Remark* 3.5 ($L_1$ Geometry vs. $L_4$ Smoothing). While AMS targets value smoothing at downstream stages ($L_{3-4}$), the geometry of the action space ($L_1$) remains a foundational factor along the Lipschitz pathway. Recent findings (Lin, 2026) suggest that action projections (e.g., hypercube versus sphere) should align with the underlying control structure of the task. From this perspective, an appropriate $L_1$ geometry can improve conditioning and reduce downstream sensitivity, while AMS remains essential for stabilizing TD updates once the action dimension is sufficiently high.

### 3.5. Practical Algorithm

We instantiate AMS within the TD3 framework (Fujimoto et al., 2018), yielding **AMS-TD3** (Figure 2). The modification is minimal: only the TD target computation changes from point evaluation to neighborhood averaging.

**Hyperparameter Guidance.** In high-dimensional action spaces, the Euclidean norm of a coordinate-wise perturbation scales as $O(\sqrt{d_a})$. Accordingly, larger neighborhoods might be required to achieve comparable geometric cov-

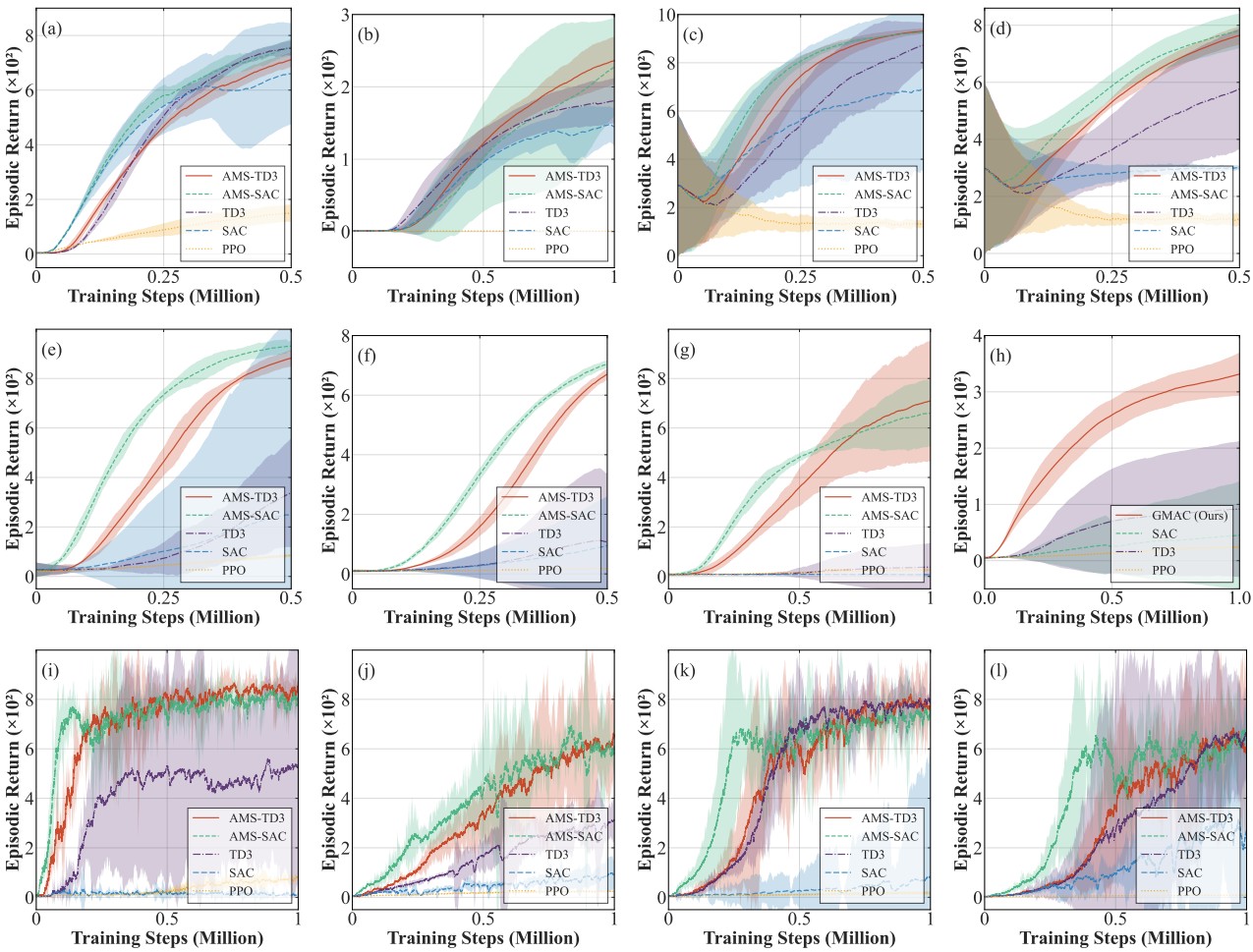

*Figure 3.* Learning curves on (a) Cheetah-Run; (b) Humanoid-Run; (c) Quadruped-Walk; (d) Quadruped-Run; (e) Dog-Stand; (f) Dog-Walk; (g) Dog-Trot; (h) Dog-Run; (i) H1-Sit; (j) H1-Slide; (k) H1-Walk; and (l) H1-Run. Shaded regions denote 95% CI.

erage as dimensionality increases. In practice, we fix the neighborhood radius to $\epsilon = 0.2$ across all benchmarks and find performance to be robust to this choice (Table 3).

**Extension to SAC.** AMS applies to SAC by replacing the point evaluation $Q(s', \tilde{a})$ with the neighborhood average centered at the policy mean $\mu(s')$. This decouples entropy exploration from value estimation, while the entropy bonus is computed from the stochastic policy as usual.

*Remark* 3.6 (Relation to Model-Based Methods). AMS and model-based methods address sample efficiency from different angles. Dreamer learns a world model for imagination-based planning and representation learning; as a byproduct, the learned dynamics $\hat{p}(s'|s, a)$ are smoother, due to supervised multi-step prediction and implicit averaging in latent dynamics rollouts, reducing $L_2$, i.e., sensitivity to action perturbations in state transitions. In contrast, AMS smooths the Q-function and TD target directly, reducing sensitivity in value estimation ($L_3$ and $L_4$). The two approaches are complementary: model-based methods constrain *upstream* sensitivity, while AMS constrains *downstream* sensitivity.

## 4. Experiments

We evaluate AMS across a diverse set of continuous control tasks to assess (i) performance on standard benchmarks, (ii) scalability to high-dimensional control, and (iii) the effectiveness of geometric smoothing across algorithm families.

### 4.1. Experimental Setup

**Environments.** We evaluate AMS on standard DMControl locomotion tasks (Tassa et al., 2018) spanning increasing action-space complexity, including the 38-D Dog domain and HumanoidBench (Sferrazza et al., 2024) (19 action dimensions on the H1 humanoid robot).

**Baselines.** We compare against SAC, TD3, and PPO (Schulman et al., 2017) using standard CleanRL (Huang et al., 2022) implementations without architectural modifications.

*Table 2.* Performance overview on DMC and HumanoidBench tasks. Mean episodic return $\pm$ standard deviation over 5 seeds.

| Method | Cheetah | Human-R | Quad-W | Quad-R | Dog-S | Dog-W | Dog-T | Dog-R | H1-Sit | H1-Slide | H1-Walk | H1-Run |
|---|---|---|---|---|---|---|---|---|---|---|---|---|
| PPO | $150_{\pm 32}$ | $1_{\pm 0}$ | $131_{\pm 18}$ | $119_{\pm 22}$ | $128_{\pm 5}$ | $18_{\pm 1}$ | $27_{\pm 6}$ | $24_{\pm 2}$ | $80_{\pm 6}$ | $24_{\pm 3}$ | $17_{\pm 7}$ | $9_{\pm 2}$ |
| SAC | $661_{\pm 185}$ | $145_{\pm 26}$ | $691_{\pm 336}$ | $301_{\pm 8}$ | $249_{\pm 781}$ | $95_{\pm 166}$ | $8_{\pm 3}$ | $45_{\pm 95}$ | $11_{\pm 4}$ | $92_{\pm 57}$ | $81_{\pm 501}$ | $237_{\pm 458}$ |
| TD3 | $753_{\pm 27}$ | $181_{\pm 31}$ | $873_{\pm 94}$ | $576_{\pm 212}$ | $337_{\pm 220}$ | $107_{\pm 230}$ | $38_{\pm 97}$ | $92_{\pm 121}$ | $527_{\pm 476}$ | $315_{\pm 92}$ | $\mathbf{787}_{\pm 57}$ | $658_{\pm 190}$ |
| **AMS-TD3** | $712_{\pm 27}$ | $\mathbf{236}_{\pm 33}$ | $\mathbf{932}_{\pm 10}$ | $765_{\pm 32}$ | $\mathbf{957}_{\pm 33}$ | $670_{\pm 18}$ | $708_{\pm 244}$ | $\mathbf{387}_{\pm 32}$ | $\mathbf{855}_{\pm 22}$ | $577_{\pm 141}$ | $775_{\pm 121}$ | $614_{\pm 278}$ |
| **AMS-SAC** | $\mathbf{757}_{\pm 31}$ | $227_{\pm 68}$ | $927_{\pm 3}$ | $\mathbf{781}_{\pm 61}$ | $931_{\pm 21}$ | $\mathbf{704}_{\pm 12}$ | $\mathbf{761}_{\pm 137}$ | $427_{\pm 25}$ | $781_{\pm 2}$ | $\mathbf{577}_{\pm 140}$ | $752_{\pm 18}$ | $\mathbf{625}_{\pm 140}$ |

AMS adds NTD and LN-Q; their individual contributions are analyzed in Section 4.3.

**Protocol.** All algorithms use 8 parallel environments and are trained for 1M steps on Humanoid-Run and Dog-Run, and 500K steps on all other tasks. We use identical hyperparameters across all environments. We report mean episodic returns $\pm$ standard deviation over 5 seeds.

**Design Choice: Isolating $L_{3-4}$ from $L_1$.** AMS-TD3 removes the Gaussian target action noise used in standard TD3, instantiating a pure $L_{3-4}$ configuration, while AMS-SAC covers $L_1 + L_{3-4}$ by retaining stochastic action sampling. This comparison tests a core Lipschitz-pathway prediction: *controlling more stages yields stronger stability*.

## 4.2. Main Results

**Low-to-Medium Dimensional Control.** On Cheetah-Run (6D) and Humanoid-Run (21D), both vanilla baselines and AMS variants achieve reasonable performance (Figure 3, Table 2). AMS provides modest but consistent gains: AMS-SAC improves SAC by 18% on Cheetah and 56% on Humanoid-Run. Notably, variance reduction is substantial. AMS-TD3 achieves comparable mean performance to TD3 on Cheetah with similar variance across seeds. These results confirm that AMS does not harm performance on simpler tasks where the discrete–continuous mismatch is less severe.

**High-Dimensional Control: Quadruped.** The benefits of geometric smoothing become pronounced on Quadruped (12D action space). As shown in Figure 3(c-d), vanilla TD3 and SAC exhibit high variance and inconsistent convergence, with SAC achieving only $301 \pm 8$ on Quadruped-Run. In contrast, AMS-SAC reaches $781 \pm 61$, representing a **2.6×** **improvement**. Learning curves show more stable training dynamics with reduced oscillation. Figure 4 provides direct mechanistic evidence: AMS-TD3 yields $\sim$2.4× smaller critic gradient norms and $\sim$4.5× smaller TD target variance than vanilla TD3 on QUADRUPED-RUN, confirming that the gains arise from downstream geometric stabilization ($L_{3\text{-}4}$) rather than from incidental optimization effects.

**Extreme High-Dimensional Control: Dog.** The Dog domain (38D action space) represents the critical test of

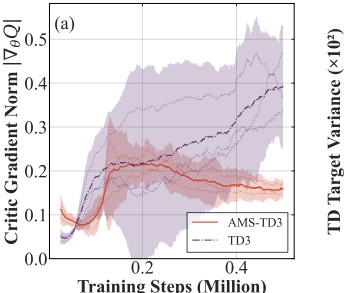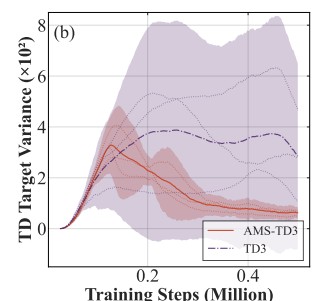

*Figure 4.* Geometric stability diagnostics on QUADRUPED-RUN. (a) Critic gradient norm $\|\nabla_\theta Q\|$ ($L_4$). (b) TD target variance $\mathrm{Var}(y)$ ($L_3$). AMS-TD3 yields $\sim$2.4× smaller gradients and $\sim$4.5× smaller target variance, confirming downstream stabilization. Mean over 3 seeds, shaded regions denote 95% CI.

our framework. Table 2 shows that vanilla algorithms fail catastrophically. TD3 achieves only $92 \pm 121$ on Dog-Run, with variance exceeding the mean, indicating that most seeds fail entirely. SAC fares no better at $45 \pm 95$, despite its entropy regularization. AMS transforms these failures into successes. AMS-SAC achieves $\mathbf{427 \pm 25}$ on Dog-Run and $\mathbf{704 \pm 13}$ on Dog-Walk, representing improvements of **9.5×** and **7.4×** respectively. Figure 3(e-h) illustrates this dramatic difference: while baselines remain flat near zero, AMS variants show steady, consistent improvement throughout training. The low standard deviations across seeds ($\pm 25$ and $\pm 13$) demonstrate that AMS provides reliable learning rather than occasional lucky runs.

**Humanoid Control: HumanoidBench.** We further evaluate on the HumanoidBench (H1) suite (Sferrazza et al., 2024), covering whole-body humanoid behaviors of varying difficulty. AMS variants match or outperform baselines on all four tasks (Table 2). On H1-Sit, AMS-TD3 achieves $855 \pm 22$ versus TD3's $527 \pm 476$, a **62%** mean improvement with > **20×** variance reduction. On the more dynamic H1-Slide, AMS-SAC reaches $577 \pm 140$, a **6.3×** improvement over SAC ($92 \pm 57$). TD3 retains a slight edge on H1-Walk and H1-Run, suggesting that for tasks where vanilla TD3 already converges stably, AMS provides comparable rather than superior performance—consistent with the Lipschitz-pathway view that $L_{3\text{-}4}$ smoothing helps most when downstream geometry is the limiting factor.

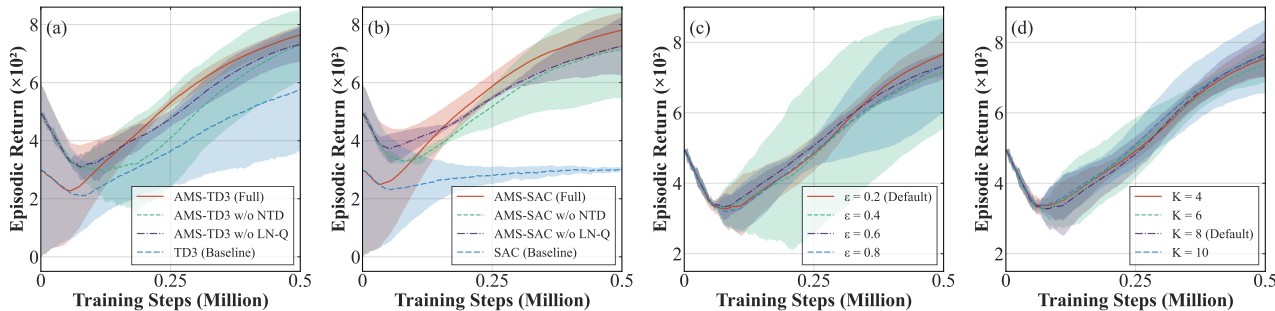

*Figure 5.* **Ablation studies on Quadruped-Run.** (a-b) Component analysis: Both NTD and LN-Q are necessary for peak performance and variance reduction in TD3 and SAC. (c-d) Robustness: Performance remains stable across a wide range of neighborhood radii $\epsilon$ and sample counts $K$, with $K = 8$ and $\epsilon = 0.2$ providing a reliable default. All results are averaged over 3 seeds $\{0, 42, 123\}$.

**Entropy Does Not Imply Geometric Stability.** Our results directly validate the design choice of isolating $L_{3-4}$ from $L_1$. Despite SAC's entropy regularization, which constrains $L_1$ (policy diversity), both SAC and TD3 fail catastrophically on Dog tasks, indicating that entropy alone does not control Q-function geometry ($L_3$). Once AMS is applied, both algorithms succeed, demonstrating that stabilizing downstream stages ($L_{3-4}$) is necessary for high-dimensional control. Moreover, AMS-SAC slightly outperforms AMS-TD3 on Dog-Run (427 vs. 387), confirming the Lipschitz-pathway prediction that controlling more stages yields stronger stability. Together, these results reveal a key insight of our framework: **exploration and geometric stability are orthogonal**, and effective high-dimensional control requires addressing both.

**Diagnostic Evidence for Downstream Stabilization.** To verify that AMS operates on the predicted $L_{3-4}$ stages, we report two diagnostic quantities on QUADRUPED-RUN (Figure 4): the critic gradient norm $\|\nabla_\theta Q\|$ and the TD target variance $\mathrm{Var}(y)$. AMS-TD3 attains a final critic gradient norm of 0.16 versus 0.39 for vanilla TD3 ($\sim 2.4\times$ reduction), and a TD target variance of 63 versus 285 ($\sim 4.5\times$ reduction). These results directly evidence the Lipschitz-pathway prediction: neighborhood-averaged Q-evaluation smooths the value landscape ($L_3$), which in turn dampens gradient magnitudes ($L_4$). The correspondence between diagnostic stabilization and downstream task performance substantiates AMS as a principled $L_{3-4}$ controller rather than an empirical trick.

### 4.3. Ablation Studies

We conduct ablation experiments on Quadruped-Run to isolate the contributions of each component and analyze hyperparameter sensitivity. Table 3 summarizes quantitative results; Figure 5 shows learning curves.

**Component Analysis.** We ablate the two core components: Neighborhood TD (NTD, targeting $L_4$) and Layer-

*Table 3.* **Ablation Study on Quadruped-Run.** We analyze the contribution of each geometric component (top) and the sensitivity to hyperparameters using the TD3 backend (bottom). Results are reported as mean $\pm$ std $\{0, 42, 123\}$.

| Category | Variant | Quadruped-Run (Score) |
|---|---|---|
| ***Component Effectiveness (TD3 / SAC)*** | | |
| *Ablation* | Baseline | $576 \pm 212$ / $301 \pm 8$ |
| | w/o NTD | $730 \pm 119$ / $714 \pm 67$ |
| | w/o LN-Q | $732 \pm 59$ / $726 \pm 41$ |
| | **AMS (Full)** | $\mathbf{765 \pm 32}$ / $\mathbf{781 \pm 61}$ |
| ***Hyperparameter Sensitivity (AMS-TD3)*** | | |
| *Radius $\epsilon$* | $\epsilon = 0.20$ **(Default)** | $\mathbf{766 \pm 26}$ |
| | $\epsilon = 0.40$ | $716 \pm 64$ |
| | $\epsilon = 0.60$ | $733 \pm 15$ |
| | $\epsilon = 0.80$ | $734 \pm 53$ |
| *Neighbors $K$* | $K = 4$ | $754 \pm 21$ |
| | $K = 6$ | $737 \pm 20$ |
| | $K = 8$ **(Default)** | $\mathbf{766 \pm 26}$ |
| | $K = 10$ | $760 \pm 42$ |

Norm in Q-networks (LN-Q, targeting $L_3$). As shown in Figure 5(a-b), neither component alone matches the full method, and their combination yields the best performance.

For TD3 (Figure 5a), the baseline achieves $576 \pm 212$, exhibiting high variance across seeds, where some seeds learn reasonable gaits while others collapse entirely. Adding LN-Q alone (w/o NTD) improves stability, reaching $730 \pm 119$, by constraining Q-network gradients. Adding NTD alone (w/o LN-Q) yields similar gains at $732 \pm 59$, by smoothing the TD target directly. The full AMS-TD3 achieves $765 \pm 32$, the highest mean with the lowest variance, confirming that both mechanisms contribute distinctly.

For SAC (Figure 5b), the pattern is even more striking. The baseline achieves only $301 \pm 8$, consistently trapped in a suboptimal region despite entropy-driven exploration. Both ablated variants (w/o NTD: $714 \pm 67$; w/o LN-Q: $726 \pm 41$) successfully escape this trap, confirming that *either* geometric constraint provides substantial benefit over entropy regularization alone. The full AMS-SAC reaches

$781 \pm 61$, demonstrating that the two components provide complementary regularization along the Lipschitz pathway.

**Synergy Between $L_3$ and $L_4$ Constraints.** A key observation is that the combination outperforms either component alone, consistent with the **multiplicative** structure of our Lipschitz pathway (Eq. 1). LN-Q constrains Q-network curvature ($L_3$), ensuring no sharp gradients amplify perturbations globally. NTD smooths the TD target ($L_4$), filtering local noise before they corrupt the Bellman backup. When both stages are constrained, perturbations are suppressed at two successive points in the learning pipeline, yielding compounded stability gains. This explains why neither technique alone fully solves high-dimensional control: controlling a single $L_i$ factor leaves others free to amplify errors.

**Robustness to Hyperparameters.** A practical concern for any new method is sensitivity to hyperparameter tuning. We vary the neighborhood radius $\epsilon$ and the number of orthogonal samples $K$ (Figure 5(c-d), Table 3).

*Neighborhood radius.* Performance is robust across $\epsilon \in \{0.2, 0.4, 0.6, 0.8\}$, with $\epsilon = 0.2$ achieving the best result ($766 \pm 26$). Larger radii introduce modest bias but remain effective, confirming that AMS is not sensitive to precise tuning of this parameter.

*Number of neighbors.* Results are stable across $K \in \{4, 6, 8, 10\}$, with $K = 8$ providing a slight edge ($766 \pm 26$). This confirms that a small number of orthogonal directions suffices to approximate the local manifold integral, as predicted by our variance reduction analysis (Theorem 3.2).

**Variance Reduction as a Stability Indicator.** Beyond mean performance, AMS dramatically reduces variance across seeds. The baseline TD3 exhibits $\pm 212$ standard deviation on Quadruped-Run, indicating that learning success depends heavily on random initialization. The full AMS-TD3 reduces this to $\pm 32$, corresponding to a **6.6×** **reduction in variance**. This demonstrates that geometric smoothing transforms high-dimensional RL from a "lucky seed" regime into reliable, reproducible learning, a critical property for practical deployment. Notably, while baseline SAC collapses on Quadruped-Run tasks, AMS-SAC achieves stable high-performance convergence at approximately 781 reward, confirming that combining $L_1$ exploration with $L_{3-4}$ smoothing enables robust convergence where entropy alone fails.

## 5. Conclusion

We address why model-free RL struggles in high-dimensional continuous control. The root cause is a *discrete–continuous mismatch*: learning continuous value landscapes from pointwise supervision yields unreliable extrapolation.

We propose **Action Manifold Smoothing (AMS)**, which stabilizes learning by replacing point TD targets with orthogonally-sampled neighborhood averages. We show this is equivalent to implicit Laplacian smoothing, suppressing spurious high-frequency oscillations while preserving robust structure. AMS is algorithm-agnostic, improving both TD3 and SAC. **This confirms a critical distinction: exploration (entropy) and geometric stability (curvature) are orthogonal concerns, and AMS resolves the latter where entropy fails.**

Empirically, AMS enables TD3 to achieve over 400 reward on Dog Run (38-dimensional actions) within one million environment steps, where vanilla algorithms completely fail. Our Lipschitz pathway framework ($L_1$–$L_4$) provides diagnostic guidance, explaining the complementary roles of action geometry ($L_1$), world models ($L_2$), and value smoothing ($L_{3-4}$). From this perspective, model-based methods smooth state transitions ($L_2$), while AMS smooths value geometry ($L_{3-4}$), addressing the same mismatch from complementary ends of the learning pipeline.

**Limitations.** While AMS effectively stabilizes high-dimensional control, it currently employs a fixed isotropic integration kernel. Extending this to **geometry-aware anisotropic smoothing**, in which the neighborhood adapts to the local curvature of the value landscape, is a promising direction. More broadly, a unified analysis of how $L_1$–$L_4$ constraints interact across different action dimensionalities remains an open theoretical question.

**Broader Impact.** AMS demonstrates that the bottleneck in high-dimensional control is not algorithmic complexity, but *geometric mismatch*. By respecting the geometry of value learning, simple principles can outperform increasingly complex machinery. We hope this geometric perspective inspires future work on principled stabilization beyond model-free RL.

*The Q-function is not a lookup table to memorize, but a landscape to smooth. When geometry is respected, action control becomes not just tractable, but principled.*

## Impact Statement

This paper aims to advance machine learning. We do not identify specific societal consequences requiring further discussion.

## Acknowledgements

This work was supported in part by the China Scholarship Council Ph.D. Scholarship for 2023–2027 (No. 202206170011). The authors thank the anonymous reviewers for their constructive feedback.

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

# A. Theoretical Analysis

In this section, we provide theoretical foundations for Action Manifold Smoothing (AMS). We first define the neighborhood target in its continuous integral form, establishing its connection to manifold smoothing, then analyze the properties of its sample-based approximation.

## A.1. Proof of Proposition 3.1

In this section, we provide the complete proof for Proposition 3.1 stated in the main text, explicitly mapping the derivation to our $L_1$–$L_4$ framework.

*Proof.* Consider two actions $a_1, a_2 \in \mathcal{A}$ applied at the same state $s$. For clarity, we assume deterministic dynamics $s' = f(s, a)$; the stochastic case follows by taking expectations. Let $s'_1 = f(s, a_1)$ and $s'_2 = f(s, a_2)$ denote the resulting next states. We introduce the following Lipschitz constants, mapped to our $L_1$–$L_4$ framework:

1. **Dynamics ($L_2$):** $\|s'_1 - s'_2\| \leq L_{\text{dyn}}\|a_1 - a_2\|$, where $L_{\text{dyn}}$ quantifies environment sensitivity.

2. **Policy ($L_1$):** The target policy $\pi$ is $L_\pi$-Lipschitz, i.e., $\|\pi(s_1) - \pi(s_2)\| \leq L_\pi\|s_1 - s_2\|$.

3. **Q-Function ($L_3$):** Critic satisfies $|Q(s_1, a) - Q(s_2, a)| \leq L_{Q,s}\|s_1 - s_2\|$ and $|Q(s, a_1) - Q(s, a_2)| \leq L_{Q,a}\|a_1 - a_2\|$.

We bound the target difference $|y(a_1) - y(a_2)|$ using the triangle inequality. Omitting the reward term (which contributes an additive constant independent of action), the variation is dominated by the value term:

$$
\begin{aligned}
\Delta y &= \gamma \left| Q(s'_1, \pi(s'_1)) - Q(s'_2, \pi(s'_2)) \right| \\
&\leq \gamma \underbrace{\left| Q(s'_1, \pi(s'_1)) - Q(s'_2, \pi(s'_1)) \right|}_{\text{(A) Sensitivity to State}} + \gamma \underbrace{\left| Q(s'_2, \pi(s'_1)) - Q(s'_2, \pi(s'_2)) \right|}_{\text{(B) Sensitivity to Action via Policy}}
\end{aligned}
\tag{10}
$$

**Term (A): Direct State Sensitivity.** Using the Lipschitz constant of $Q$ w.r.t state:

$$
\text{(A)} \leq L_{Q,s}\|s'_1 - s'_2\| \leq L_{Q,s}L_{\text{dyn}}\|a_1 - a_2\|.
\tag{11}
$$

**Term (B): Coupled Action Sensitivity.** Using the Lipschitz constants of $Q$ (w.r.t action) and $\pi$:

$$
\text{(B)} \leq L_{Q,a}\|\pi(s'_1) - \pi(s'_2)\| \leq L_{Q,a}L_\pi\|s'_1 - s'_2\| \leq L_{Q,a}L_\pi L_{\text{dyn}}\|a_1 - a_2\|.
\tag{12}
$$

**Total Sensitivity.** Summing the terms and factoring out $\|a_1 - a_2\|$:

$$
|y(a_1) - y(a_2)| \leq \gamma \left( L_{Q,s} + L_{Q,a}L_\pi \right) L_{\text{dyn}}\|a_1 - a_2\|.
\tag{13}
$$

Thus, the effective Lipschitz constant of the TD target satisfies

$$
L_{\text{target}} = \gamma L_{\text{dyn}}(L_{Q,s} + L_{Q,a}L_\pi) = O(L_2 \cdot L_3),
\tag{14}
$$

which completes the proof. $\square$

*Remark* A.1 (Connection to Unified Framework). This derivation mathematically grounds the intuition in Eq. 1.

- $L_{\text{dyn}}$ **corresponds to** $L_2$: In chaotic environments (e.g., Dog Run), $L_{\text{dyn}}$ is large, amplifying input perturbations.

- $(L_{Q,s} + L_{Q,a}L_\pi)$ **corresponds to** $L_3$: This composite term represents the total sensitivity of the Critic. It reveals why spectral normalization on $Q$ (constraining $L_{Q,s}, L_{Q,a}$) helps, but also why it depends on the policy's smoothness $L_\pi$. Here $L_\pi$ characterizes sensitivity of the policy to *state* perturbations, and should not be confused with $L_1$ in Eq. (1), which refers to the geometry of *action parameterization*. When the policy varies slowly, this composite sensitivity is dominated by $L_{Q,s}$, so the effective amplification primarily scales with $L_{Q,s}L_{\text{dyn}}$.

- $L_4$ **(TD target stability)**: While $L_3$ measures Q-network smoothness, $L_4$ captures sensitivity in the target *computation* itself. Standard point evaluation $Q(s', \pi(s'))$ corresponds to a Dirac measure in action space, whereas NTD replaces it with a local integral, reducing the effective Lipschitz constant of the target computation.

- **Multiplicative Amplification**: The product $L_2 \times L_3 \times L_4$ confirms that instability compounds across stages. AMS jointly reduces $L_3$ (via implicit Laplacian regularization) and $L_4$ (via neighborhood averaging).

## A.2. Continuous Formulation: TD as Manifold Convolution

We provide the theoretical foundation for NTD, establishing its continuous formulation as manifold convolution. The detailed proofs follow in subsequent sections.

Standard TD learning evaluates the Q-function at a single point:

$$y_{\text{td}} = r + \gamma\, Q(s', \mu'), \quad \text{where } \mu' = \pi(s'). \tag{15}$$

We replace this point evaluation with a **local expectation** over the action manifold. Let $\mathcal{B}_\epsilon(\mu')$ denote an $\epsilon$-neighborhood around $\mu' = \pi(s')$, and let $p_{\text{geo}}(a|\mu')$ be a geometry-aware smoothing kernel. The **NTD target** is:

$$y_{\text{ntd}} = r + \gamma \int_{\mathcal{A}} Q(s', a)\, p_{\text{geo}}(a|\mu')\, da \tag{16}$$

This integral has a natural interpretation as **convolution**: the Q-function is convolved with the smoothing kernel $p_{\text{geo}}$, filtering out high-frequency artifacts (spurious peaks) while preserving the dominant low-frequency structure of the value landscape.

**Connection to Laplacian smoothing.** When $p_{\text{geo}}$ is the uniform distribution on the sphere $\partial\mathcal{B}_\epsilon(\mu')$ (i.e., $a = \mu' + \epsilon \mathbf{u}$ with $\mathbf{u} \sim \text{Uniform}(\mathbb{S}^{d_a-1})$), we obtain via Taylor expansion:

$$\mathbb{E}_{a \sim \mathcal{B}_\epsilon(\mu')}[Q(s', a)] = Q(s', \mu') + \frac{\epsilon^2}{2d_a} \Delta_a Q(s', \mu') + O(\epsilon^4) \tag{17}$$

where $\Delta_a Q = \text{tr}(\nabla_a^2 Q)$ is the Laplacian. This is precisely the result of Theorem 3.3; the full derivation is provided in Appendix A.4.

**Discrete approximation.** In practice, we approximate via Monte Carlo with $K$ samples:

$$y_{\text{ntd}} \approx r + \gamma \cdot \frac{1}{K} \sum_{k=1}^{K} Q(s', a_k'), \quad a_k' \sim p_{\text{geo}}(\cdot \,|\, \mu') \tag{18}$$

To maximize coverage efficiency, we use orthogonal sampling directions (Section A.5) rather than i.i.d. Gaussian noise.

*Remark* A.2 (Geometric Interpretation). NTD acts as a **geometric low-pass filter**:

- $\Delta_a Q > 0$ (local minimum): neighborhood average **increases** target, filling valleys.
- $\Delta_a Q < 0$ (local maximum): neighborhood average **decreases** target, smoothing peaks.

This is mathematically equivalent to evolving $Q$ under the heat equation $\partial_t Q = \Delta Q$ for time $O(\epsilon^2)$, a classical smoothing operation in differential geometry. This directly constrains $L_3$ in our Lipschitz pathway by regularizing high-frequency artifacts.

## A.3. Proof of Theorem 3.2 (Variance Reduction)

We show that NTD yields more stable targets than point TD by suppressing variance induced by local irregularities in the Q-function.

*Proof.* Let $\mu' = \pi(s')$ be the anchor action and $\{a_k'\}_{k=1}^K$ be samples in an $\epsilon$-neighborhood around $\mu'$. Define

$$y_{\text{td}} = r + \gamma Q(s', \mu'), \quad y_{\text{ntd}} = r + \gamma \cdot \frac{1}{K} \sum_{k=1}^{K} Q(s', a_k'). \tag{19}$$

**Step 1: Variance decomposition.** Since the reward $r$ is constant given $(s, a)$,

$$\text{Var}[y_{\text{ntd}}] = \gamma^2 \, \text{Var}\left[\frac{1}{K} \sum_{k=1}^{K} Q(s', a_k')\right]. \tag{20}$$

**Step 2: Lipschitz control of perturbation effects.** Assume that $Q(s', \cdot)$ is $L$-Lipschitz in action. Then

$$|Q(s', a'_k) - Q(s', \mu')| \leq L\|a'_k - \mu'\| \leq L\epsilon, \tag{21}$$

so all sampled Q-values lie in $[Q(s', \mu') - L\epsilon, \ Q(s', \mu') + L\epsilon]$.

**Step 3: Variance bound via Popoviciu's inequality.** By Lipschitz continuity, all sampled Q-values lie in an interval of width $2L\epsilon$. Popoviciu's inequality states that $\mathrm{Var}[X] \leq (b-a)^2/4$ for $X \in [a, b]$. Thus, for a single sample:

$$\mathrm{Var}[Q(s', a'_k)] \leq \frac{(2L\epsilon)^2}{4} = L^2\epsilon^2. \tag{22}$$

For $K$ i.i.d. samples, the variance of the average is reduced by $K$:

$$\mathrm{Var}\left[\frac{1}{K}\sum_{k=1}^{K} Q(s', a'_k)\right] \leq \frac{L^2\epsilon^2}{K}. \tag{23}$$

**Step 4: Net variance reduction.** While neighborhood averaging introduces a bounded perturbation term $O(L^2\epsilon^2/K)$, it suppresses variance caused by sharp local irregularities and spurious extrema in the Q-landscape. When such geometric artifacts dominate, as is common in high-dimensional action spaces, the smoothing effect outweighs the perturbation cost, yielding more stable TD targets.

$\square$

*Remark* A.3 (Orthogonal Sampling Advantage). The above analysis assumes i.i.d. samples. AMS employs orthogonal sampling, which maximizes the spanned neighborhood volume and avoids redundant probing. By ensuring $\mathbf{v}_i^\top \mathbf{v}_j = 0$ for $i \neq j$, orthogonal directions reduce covariance between samples, providing further variance reduction beyond the $1/K$ averaging effect (see Appendix A.5).

## A.4. Proof of Theorem 3.3 (Implicit Hessian Regularization)

We derive the second-order expansion of the neighborhood expectation under uniform sampling on the spherical surface $\partial\mathcal{B}_\epsilon(\mu')$. The discrete $K$-sample orthogonal scheme approximates this continuous integral.

*Proof.* Let $a' = \mu' + \epsilon\mathbf{u}$, where $\mathbf{u}$ is drawn uniformly from the unit sphere $\mathbb{S}^{d_a-1}$. Expand $Q(s', a')$ around $\mu'$, with gradient and Hessian evaluated at $(s', \mu')$:

$$Q(s', a') = Q(s', \mu') + \epsilon\nabla_a Q^\top \mathbf{u} + \frac{\epsilon^2}{2}\mathbf{u}^\top \mathbf{H}\mathbf{u} + O(\epsilon^3), \tag{24}$$

where $\mathbf{H} = \nabla_a^2 Q(s', \mu')$ is the Hessian matrix.

**Step 1: First-order term.** By spherical symmetry, $\mathbb{E}[\mathbf{u}] = \mathbf{0}$, so:

$$\mathbb{E}[\epsilon\nabla_a Q^\top \mathbf{u}] = 0. \tag{25}$$

**Step 2: Second-order term.** For uniform distribution on $\mathbb{S}^{d-1}$, standard results in high-dimensional probability (Vershynin, 2018) yield:

$$\mathbb{E}[\mathbf{u}\mathbf{u}^\top] = \frac{1}{d_a}\mathbf{I}. \tag{26}$$

Therefore:

$$\mathbb{E}\left[\frac{\epsilon^2}{2}\mathbf{u}^\top \mathbf{H}\mathbf{u}\right] = \frac{\epsilon^2}{2}\mathrm{tr}\left(\mathbf{H} \cdot \frac{1}{d_a}\mathbf{I}\right) = \frac{\epsilon^2}{2d_a}\mathrm{tr}(\mathbf{H}) = \frac{\epsilon^2}{2d_a}\Delta_a Q. \tag{27}$$

**Step 3: Higher-order terms.** Odd-order terms vanish by symmetry ($\mathbf{u} \to -\mathbf{u}$ invariance). The residual is $O(\epsilon^4)$.

**Step 4: Final result.**

$$\mathbb{E}[y_{\mathrm{ntd}}] \approx r + \gamma\left(Q(s', \mu') + \frac{\epsilon^2}{2d_a}\Delta_a Q + O(\epsilon^4)\right). \tag{28}$$

$\square$

*Remark* A.4 (Regularization Effect). The correction term $\frac{\epsilon^2}{2d_a}\Delta_a Q$ acts as implicit curvature regularization. At local maxima (peaks), $\Delta_a Q < 0$, so the target is reduced; at local minima (valleys), $\Delta_a Q > 0$, so the target is increased. Training Q-networks to match these smoothed targets implicitly penalizes high curvature, constraining $L_3$ in our Lipschitz pathway in Eq. (1).

## A.5. Analysis of Orthogonal vs Random Sampling

We analyze why orthogonal sampling provides superior efficiency compared to independent random sampling for NTD estimation.

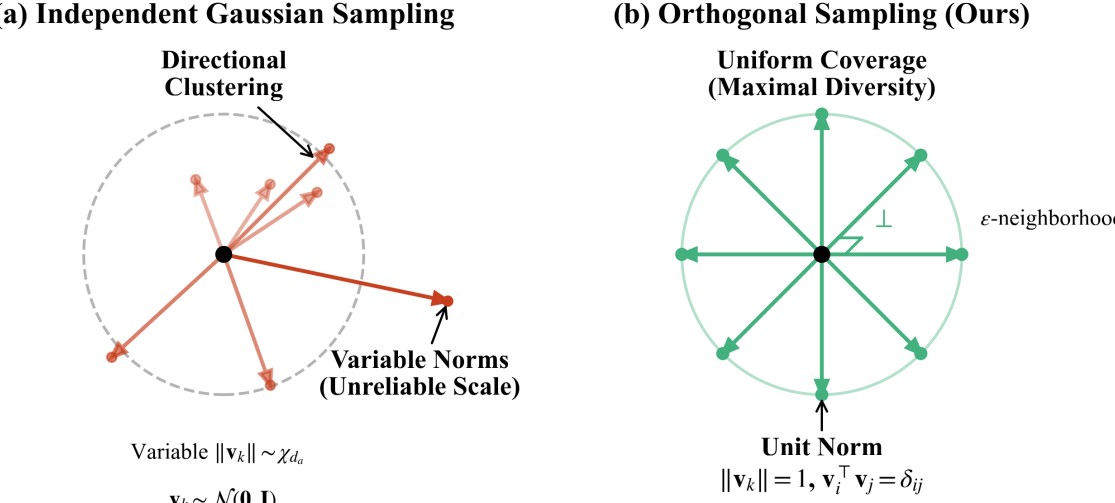

**(a) Independent Gaussian Sampling**

Directional Clustering

Variable Norms (Unreliable Scale)

Variable $\|\mathbf{v}_k\| \sim \chi_{d_a}$

$\mathbf{v}_k \sim \mathcal{N}(\mathbf{0}, \mathbf{I})$

*Problem: Redundant directions & inconsistent magnitudes*

**(b) Orthogonal Sampling (Ours)**

Uniform Coverage (Maximal Diversity)

$\perp$

$\varepsilon$-neighborhood

Unit Norm

$\|\mathbf{v}_k\| = 1$, $\mathbf{v}_i^\top \mathbf{v}_j = \delta_{ij}$

*Benefit: Maximal volume coverage & zero cross-covariance*

*Figure A.1.* **Comparison of sampling strategies for NTD.** (a) Independent Gaussian sampling exhibits directional clustering and variable magnitudes, leading to redundant coverage. (b) Orthogonal sampling via QR decomposition enforces unit norm and mutual orthogonality ($\mathbf{v}_i^\top \mathbf{v}_j = \delta_{ij}$), maximizing subspace coverage with minimal samples.

**Geometric Motivation.** Consider $K$ perturbation directions used to sample the neighborhood around anchor action $\mu' = \pi(s')$. Each direction $\mathbf{v}_k$ produces a point $a'_k = \mu' + \epsilon\mathbf{v}_k$ where we evaluate the Q-function. The collection of these points should ideally cover the local neighborhood as uniformly as possible, avoiding redundant samples that waste computational budget.

Independent Gaussian sampling $\mathbf{v}_k \sim \mathcal{N}(\mathbf{0}, \mathbf{I}_{d_a})$ provides no coordination between samples. While high-dimensional Gaussian vectors are approximately orthogonal in expectation, finite batches exhibit non-negligible correlations. More critically, the magnitude $\|\mathbf{v}_k\|_2$ varies according to a $\chi_{d_a}$ distribution, causing inconsistent perturbation scales across samples.

**Orthogonal Noise Construction.** Figure A.1 illustrates the contrast between random and orthogonal sampling. We construct orthogonal directions through QR decomposition. Given $K \leq d_a$ samples and action dimension $d_a$, we sample a Gaussian matrix $\mathbf{G} \in \mathbb{R}^{d_a \times K}$ with independent $\mathcal{N}(0, 1)$ entries and compute:

$$\mathbf{G} = \mathbf{Q}\mathbf{R}, \tag{29}$$

where $\mathbf{Q} \in \mathbb{R}^{d_a \times K}$ has $K$ orthonormal columns. The orthogonalized directions are the columns of $\mathbf{Q}$, each with unit norm $\|\mathbf{v}_k\|_2 = 1$.

**Volume Maximization.** The advantage of orthogonal sampling can be quantified through the volume of the parallelepiped spanned by the sample directions.

**Proposition A.5** (Orthogonal Vectors Maximize Spanned Volume)**.** *Let* $\mathbf{v}_1, \ldots, \mathbf{v}_K \in \mathbb{R}^{d_a}$ *with* $K \leq d_a$ *be vectors with fixed norm* $\|\mathbf{v}_k\|_2 = 1$*. The* $K$*-dimensional volume of the parallelepiped spanned by these vectors is:*

$$V_K = \sqrt{\det(\mathbf{V}\mathbf{V}^\top)}, \tag{30}$$

*where* $\mathbf{V} = [\mathbf{v}_1, \ldots, \mathbf{v}_K]^\top \in \mathbb{R}^{K \times d_a}$. *This volume is maximized when* $\mathbf{v}_i^\top \mathbf{v}_j = 0$ *for all* $i \neq j$.

*Proof.* The Gram matrix $\mathbf{V}\mathbf{V}^\top \in \mathbb{R}^{K \times K}$ is positive semi-definite with diagonal entries $(\mathbf{V}\mathbf{V}^\top)_{ii} = \|\mathbf{v}_i\|_2^2 = 1$ and off-diagonal entries $(\mathbf{V}\mathbf{V}^\top)_{ij} = \mathbf{v}_i^\top \mathbf{v}_j$. By Hadamard's inequality (Horn & Johnson, 2012) for positive semi-definite matrices:

$$\det(\mathbf{V}\mathbf{V}^\top) \leq \prod_{i=1}^{K} (\mathbf{V}\mathbf{V}^\top)_{ii} = 1. \tag{31}$$

Equality holds if and only if the Gram matrix is diagonal, which occurs precisely when $\mathbf{v}_i^\top \mathbf{v}_j = 0$ for all $i \neq j$. For orthogonal unit vectors, $V_K = 1$, while correlated vectors yield strictly smaller volume. $\qquad\square$

This result establishes that orthogonal sampling provides maximal directional diversity. In the context of NTD, orthogonal directions explore maximally distinct regions of the local action manifold, reducing redundancy in Q-function evaluation.

**Variance Reduction in Q-Estimation.** Beyond geometric coverage, orthogonal sampling improves statistical efficiency. Consider the estimator $\hat{Q}_{\text{avg}} = \frac{1}{K} \sum_{k=1}^{K} Q(s', a'_k)$.

Under a local linear approximation $Q(s', \mu' + \epsilon\mathbf{v}) \approx Q(s', \mu') + \epsilon\,\mathbf{g}^\top \mathbf{v}$, where $\mathbf{g} = \nabla_a Q(s', \mu')$, the covariance between two samples is:

$$\text{Cov}[Q(s', a'_i), Q(s', a'_j)] \approx \epsilon^2 (\mathbf{g}^\top \mathbf{v}_i)(\mathbf{g}^\top \mathbf{v}_j). \tag{32}$$

Since the orthogonal basis is drawn uniformly from the orthogonal group $O(d_a)$ according to the Haar measure, the joint distribution is rotationally invariant. This symmetry implies:

$$\mathbb{E}_{\text{basis}}[\mathbf{v}_i^\top \mathbf{v}_j] = 0, \quad \forall i \neq j. \tag{33}$$

Consequently, the expected cross-covariance vanishes:

$$\mathbb{E}_{\text{basis}}[\text{Cov}[Q(s', a'_i), Q(s', a'_j)]] = 0, \quad \forall i \neq j, \tag{34}$$

unlike i.i.d. sampling where random alignment induces non-zero cross-covariance in any finite sample. This implies that, under the linear approximation, orthogonal sampling achieves variance no greater than i.i.d. sampling:

$$\mathbb{E}_{\text{basis}}[\text{Var}[\hat{Q}_{\text{avg}}^{\text{orth}}]] \leq \mathbb{E}_{\text{samples}}[\text{Var}[\hat{Q}_{\text{avg}}^{\text{iid}}]]. \tag{35}$$

The reduction is most significant when the Q-function has strong directional dependence, precisely the case in high-dimensional action spaces.

**Practical Implications.** For the Dog task with $d_a = 38$ and $K = 8$, orthogonal sampling ensures the 8 directions span an 8-dimensional subspace with maximal volume, rather than potentially clustering in a lower-dimensional region. This geometric guarantee becomes increasingly important as dimension grows, where naive random sampling rapidly loses coverage efficiency.

### A.6. Remarks on LayerNorm in Q-Networks

We briefly discuss why LayerNorm improves stability in high-dimensional RL tasks, complementing the empirical results in Section 4.3.

**Gain control property.** Layer Normalization (Ba et al., 2016) normalizes intermediate activations to have zero mean and unit variance. For an input vector $\mathbf{z} \in \mathbb{R}^d$ (e.g., a hidden layer activation), LayerNorm computes:

$$\text{LN}(\mathbf{z}) = \frac{\mathbf{z} - \bar{z}}{\sigma_z} \odot \mathbf{g} + \mathbf{b}, \tag{36}$$

where $\bar{z} = \frac{1}{d} \sum_i z_i$ is the mean, $\sigma_z = \sqrt{\frac{1}{d} \sum_i (z_i - \bar{z})^2}$ is the standard deviation, and $\mathbf{g}, \mathbf{b} \in \mathbb{R}^d$ are learnable scale and shift parameters.

This normalization provides implicit gain control: if pre-activations scale by a factor $\alpha$, the standard deviation $\sigma_z$ also scales by $\alpha$, mitigating excessive gradient amplification. Consequently, the local Lipschitz constant of each layer is effectively bounded, reducing the overall Lipschitz constant $L_Q$ of the Q-network (see Proposition 3.1). In practice, this makes $\|\nabla_a Q(s, a)\|$ less prone to extreme values, limiting sensitivity to input perturbations.

**Connection to AMS.** LayerNorm complements NTD by operating at different points in the sensitivity pathway: NTD smooths the *TD target* ($L_4$), while LayerNorm regularizes the *Q-network itself* ($L_3$). Together, they provide complementary geometric regularization in our Lipschitz framework.

**When LayerNorm helps.** Our experiments (Section 4.3) confirm that LayerNorm is most beneficial for high-dimensional tasks with strongly coupled action dynamics, where cross-dimension normalization helps coordinate sensitivity across interdependent action dimensions. LayerNorm's effectiveness depends on whether the task's action space exhibits persistent cross-dimensional coupling. When actions jointly influence the same physical degrees of freedom, global normalization stabilizes learning by preventing any single dimension from dominating the gradient signal.

For formal analyses of normalization techniques and their Lipschitz-related properties in off-policy RL, we refer readers to (Wang et al., 2025).

### A.7. Remarks on Action Space Geometry

We briefly discuss the geometric priors imposed by different action projections.

**Spherical Projection (Normalize).** Maps actions to a hypersphere:

$$a = \frac{\mu}{\|\mu\|_2} \cdot a_{\text{scale}} + a_{\text{bias}} \tag{37}$$

This enforces equal magnitude across all action dimensions, suitable for tasks requiring isotropic force distribution where no single joint should dominate.

**Cuboid Projection (Tanh).** Maps actions to a hypercube $[-1, 1]^{d_a}$:

$$a_i = \tanh(\mu_i) \cdot a_{\text{scale},i} + a_{\text{bias},i} \tag{38}$$

This allows independent saturation per dimension, suitable for tasks requiring asymmetric joint control (e.g., quadruped gait with alternating leg extension).

**Geometric interpretation.** From the Lipschitz pathway perspective, action projection defines the $L_1$ geometry of the control space and determines how action perturbations are structured before propagating through the dynamics and value function. This $L_1$ choice is complementary to the $L_{3-4}$ regularization introduced by AMS, and its optimal form depends on the underlying control structure of the task. A systematic study of projection geometry across task families is left for future work.

## Appendix B. Experimental Details

### B.1. Environment Descriptions

We evaluate AMS on six DeepMind Control Suite environments spanning a range of action dimensions. Table B.1 summarizes the key characteristics.

**Environment Selection Rationale.** We select environments to test AMS across a spectrum of action dimensionality: Cheetah-Run (6D) serves as a low-dimensional baseline to verify that AMS does not harm performance on simpler tasks; Humanoid-Run (21D) and Quadruped (12D) represent medium-to-high dimensional challenges; Dog (38D) represents the extreme high-dimensional regime where vanilla algorithms completely fail.

### B.2. Implementation Details

Our implementation is based on CleanRL (Huang et al., 2022). We use identical network architectures and hyperparameters across all environments.

*Table B.1.* Environment specifications. All environments are from DeepMind Control Suite (DMC). We select tasks with increasing action dimensionality to validate AMS across different scales.

| Environment | State Dim | Action Dim | Coupling |
|---|---|---|---|
| Cheetah-Run | 17 | 6 | Medium |
| Humanoid-Run | 67 | 21 | High |
| Quadruped-Walk | 78 | 12 | High |
| Quadruped-Run | 78 | 12 | High |
| Dog-Walk | 223 | 38 | High |
| Dog-Run | 223 | 38 | High |

**Network Architecture.** Both Actor and Critic use 3-layer MLPs with 256 hidden units and ReLU activations. We apply LayerNorm after each hidden layer in the Critic for all tasks.

*Table B.2.* AMS hyperparameters (fixed across all environments).

| Hyperparameter | Value |
|---|---|
| Number of orthogonal samples $K$ | 8 |
| Neighborhood radius $\epsilon$ | $0.2 \times a_{\text{scale}}$ |

**Base Algorithm Hyperparameters.** We use default CleanRL hyperparameters for TD3 and SAC without modification:

*Table B.3.* Base algorithm hyperparameters.

| Hyperparameter | TD3 | SAC |
|---|---|---|
| Learning rate (Actor) | $3 \times 10^{-4}$ | $3 \times 10^{-4}$ |
| Learning rate (Critic) | $3 \times 10^{-4}$ | $1 \times 10^{-3}$ |
| Batch size | 256 | 256 |
| Warmup steps | 25000 | 5000 |
| Replay buffer size | $10^6$ | $10^6$ |
| Discount factor $\gamma$ | 0.99 | 0.99 |
| Target update rate $\tau$ | 0.005 | 0.005 |
| Policy delay (TD3 only) | 2 | – |
| Entropy coefficient (SAC) | – | auto-tuned |
| Number of parallel environments | 8 | 8 |

**No Hyperparameter Tuning.** We emphasize that all results use default hyperparameters without environment-specific tuning. This demonstrates the robustness of AMS across diverse tasks.

### B.3. Hyperparameter Sensitivity

Detailed hyperparameter ablations (neighborhood radius $\epsilon$ and number of orthogonal samples $K$) are presented in Section 4.3 and Table 3. Here we summarize the key findings:

- **Number of samples $K$:** Performance saturates around $K = 8$, which provides sufficient coverage for action spaces up to 38 dimensions while adding minimal computational overhead.

- **Neighborhood radius $\epsilon$:** Results are robust across $\epsilon \in \{0.2, 0.4, 0.6, 0.8\}$. We use $\epsilon = 0.2 \times a_{\text{scale}}$ as the default.

### B.4. Computational Overhead

**Hardware.** All experiments were conducted on NVIDIA RTX 3090 GPUs.

**Training Time.** Approximate wall-clock time per 1M environment steps:

*Table B.4.* Training time comparison.

| Environment | Baseline | AMS |
|---|---|---|
| Cheetah-Run | $\sim$20 min | $\sim$25 min |
| Quadruped-Run/Walk | $\sim$1 hour | $\sim$1.5 hours |
| Humanoid-Run | $\sim$1.5 hours | $\sim$2 hours |
| Dog-Run/Walk | $\sim$3.5 hours | $\sim$4.5 hours |

**Overhead Analysis.** AMS introduces $K$ additional Q-network forward passes per update. We implement this efficiently via batched computation: all $K$ neighbor actions are evaluated in a single forward pass by reshaping tensors, avoiding the naive $K$-fold loop overhead. With this optimization, AMS adds approximately 25–30% training time, modest compared to the substantial performance gains in high-dimensional tasks.

**Random Seeds.** All main experiments use 5 random seeds. Ablation studies use 3 seeds.

**Reproducibility.** To ensure transparency, we provide TensorBoard training logs for all experiments across 5 random seeds in the supplementary material, together with visualization scripts to reproduce all reported figures.

