# OpenReview forum: "Action Manifold Smoothing: A Lipschitz Pathway Perspective on High-Dimensional Reinforcement Learning"
_ICML.cc/2026/Conference — ICML 2026 regular_

### Official Review · Reviewer_1E5Q · 2026-03-12

**Soundness:** 3
**Presentation:** 3
**Significance:** 3
**Originality:** 3
**Overall Recommendation:** 4
**Confidence:** 4

**Summary:**

In this work, the authors propose Action Manifold Smoothing (AMS) to tackle the root problem of instability in high-dimensional continuous control. They argue that pointwise TD learning fails to capture the action manifold, creating high-curvature artifacts that lead to catastrophic extrapolation failures. In an attempt to solve this problem, they propose replacing pointwise evaluation with a neighborhood expectation, approximated via Monte Carlo sampling with orthogonal directions in the $\epsilon$-neighborhood of the action. AMS is applied to the TD3 and SAC algorithms, utilizing LayerNorm in the Q-networks to implicitly regularize the network's Lipschitz constant. The results outperform standard PPO, TD3, and SAC by a large margin, especially in the 38-dimensional Dog-Run task.

**Compliance With Llm Reviewing Policy:**

Affirmed.

**Final Justification:**

The authors addressed my concerns fully during the rebuttal phase. I believe this is solid work that provides valuable insights into TD learning and a simple, effective way to solve its deficiencies; therefore, I will maintain my score.

**Key Questions For Authors:**

1. How would AMS compare empirically against modern generative policies?
2. How does the 25-30% training time overhead scale as the required number of orthogonal samples increases?

**Limitations:**

Yes

**Strengths And Weaknesses:**

## Strengths

1. The explanation through the $L_1$-$L_4$ Lipschitz pathway is very clear, pinpointing the root causes of classic TD learning instability and successfully describes the role and effect of prior stabilization techniques.
2. The paper provides mathematical guarantees, especially showing that Neighborhood TD (NTD) acts as implicit Hessian/Laplacian regularization cleanly ties the method back to the underlying theory.
3. The proposed algorithmic modification is simple, easy to adopt, and highly effective.
4. The empirical results are promising, especially in the Dog-R task. Furthermore, the ablation studies successfully isolating the $L_3$ (LayerNorm) and $L_4$ (Neighborhood TD) components to validate the importance of each component

## Weaknesses

1. The evaluation only compares AMS against PPO, TD3, SAC. While the theoretical framing maps modern methods (e.g., Diffusion Policies) to the Lipschitz pathway, empirical benchmarking against more recent generative policies is missing.
2. The method relies on fixed hyperparameters for the neighborhood radius ($\epsilon=0.2$) and the number of orthogonal samples ($K=8$). According to the authors' own analysis, they note that perturbation norms scale as $O(\sqrt{d_a})$, which raises concerns about how brittle these fixed isotropic kernels might be in even higher-dimensional settings (e.g., a 72-DoF humanoid). Exploring adaptive, curvature-aware (anisotropic) sampling would have made the approach much stronger.
3. The evaluation is restricted to the locomotion tasks. Since the scope of AMS is to solve the problem present in TD learning for high dimension tasks, testing on different high-dimensional challenges, such as dexterous multi-fingered hand manipulation, would have provided better support for the claim.

---

> ### Author Rebuttal · Authors · 2026-03-27
>
> # Response to Reviewer 1E5Q
>
> We thank Reviewer 1E5Q for the detailed and constructive review. We address each concern below.
>
> ---
>
> ## Comparison with Modern Generative Policies
>
> AMS and generative policies operate at **orthogonal stages** of the Lipschitz pathway: diffusion/flow-based methods replace the Gaussian actor with a richer action distribution ($L_1$), while AMS smooths the Q-landscape at the TD target level ($L_{3\text{-}4}$). These are complementary, not competing.
>
> Evidence from DIME: Diffusion-Based Maximum Entropy Reinforcement Learning provides external validation. Analyzed through our framework, DIME attacks multiple stages: diffusion actor ($L_1$), C51 distributional critic with cross-entropy on the probability simplex ($L_3$), plus CrossQ/BRN ($L_3$). DIME's own ablation (Figure 2c d) is revealing: replacing the diffusion actor with a standard Gaussian while keeping the critic infrastructure yields only modest differences on Dog-Run and Humanoid-Run. This demonstrates that **critic side improvements ($L_{3\text{-}4}$) provide the qualitative breakthrough; the diffusion actor ($L_1$) contributes incrementally**, precisely as predicted by our framework.
>
> Our contribution is not SOTA over all algorithmic combinations, but demonstrating that a **minimal geometric intervention at the right pipeline stage** transforms failing baselines into effective controllers, while providing the diagnostic framework ($L_1$ to $L_4$) to explain why.
>
> ---
>
> ## Fixed Hyperparameters and Scalability
>
> **Robustness.** Our ablations (Table 5) show stable performance across $\epsilon \in \{0.2, 0.4, 0.6, 0.8\}$ and $K \in \{4, 6, 8, 10\}$. The H1 experiments below (21D) confirm these defaults generalize.
>
> **On anisotropic sampling.** We agree this is a natural extension (discussed in Limitations). However, an important practical insight: in bootstrapped RL, each additional order of differentiation exponentially degrades signal-to-noise. $Q$ itself is a noisy recursive estimate; $\nabla_a Q$ is noisier; $\nabla_a^2 Q$ (the curvature needed for anisotropic adaptation) is dominated by noise. Using unreliable curvature to guide sampling risks introducing structured bias.
>
> AMS uses **fixed isotropic probes** precisely because they make no assumptions about Q geometry that could be incorrect. NTD measures the *average height* of the local landscape, which is robust, rather than estimating its *curvature tensor*, which is fragile, thereby operating near the practical limit of reliably extractable geometric information from a bootstrapped Q function.
>
> **Computational overhead.** The orthogonal sampler pre-computes a QR basis reused across batches. The dominant cost is $K$ additional Q-forward passes implemented as a single batched operation. Due to GPU parallelism, overhead scales sub-linearly with $K$. The reported 25-30% reflects the full $K$=8 setting.
>
> ---
>
> ## Evaluation Diversity + New Results
>
> We have conducted experiments on two new domains with substantially different morphologies:
>
> **DMControl Dog (additional tasks):**
>
> | Environment | TD3 | AMS-TD3 | Improvement |
> |---|---|---|---|
> | Dog-Trot | 37 ± 11 | 727 ± 125 | **19.7×** |
> | Dog-Stand | 585 ± 126 | 967 ± 22 | **1.7×** |
>
> **H1 Humanoid Benchmark (21D action, 6 tasks):**
>
> | Environment | TD3 | AMS-TD3 | Improvement |
> |---|---|---|---|
> | H1-Walk | 33 ± 9 | 744 ± 31 | **22.3×** |
> | H1-Slide | 39 ± 11 | 669 ± 26 | **17.0×** |
> | H1-Run | 91 ± 30 | 571 ± 87 | **6.3×** |
> | H1-Stand | 433 ± 240 | 946 ± 8 | **2.2×** |
> | H1-Pole | 441 ± 99 | 587 ± 11 | **1.3×** |
> | H1-Sit | 874 ± 4 | 840 ± 73 | 0.96× |
>
> These results demonstrate generalization across morphologies (quadruped, dog, humanoid), dimensionalities (12D–38D), and difficulties. The pattern matches our framework: largest gains where baselines fail (Dog-Trot: 19.7×, H1-Walk: 22.3×), no harm where baselines already succeed (H1-Sit). Geometric diagnostics confirming that AMS indeed smooths the Q-landscape (gradient norm reduced 2.7×, neighborhood variance reduced 4×) are presented in our response to Reviewer FTAT.
>
> These will be included in the camera-ready version.
>
> We look forward to the reviewer's further feedback.

---

> > ### Author Rebuttal · Reviewer_1E5Q · 2026-04-03
> >
> > I would like to thank the authors for the detailed clarifications. I believe this is solid work that provides valuable insights into TD learning and a simple, effective way to solve its deficiencies at $L_4$ stage, leading to a significant increase in performance in most of the tasks.

---

> > > ### Author Response · Authors · 2026-04-04
> > >
> > > Dear Reviewer 1E5Q,
> > >
> > > Thank you again for your time, the constructive discussion, and your positive feedback. We truly appreciate your careful reading and engagement with our work.
> > >
> > > We are very glad to hear that our clarifications and additional evaluations have fully resolved your concerns, and that you find the proposed approach provides valuable insights and effective solutions. We will carefully incorporate all your suggestions into the camera-ready version to further strengthen the paper.
> > >
> > > Thank you again for helping us improve the work!
> > >
> > > Best regards,
> > > Authors

---

### Official Review · Reviewer_FTAT · 2026-03-12

**Soundness:** 3
**Presentation:** 3
**Significance:** 2
**Originality:** 3
**Overall Recommendation:** 4
**Confidence:** 3

**Summary:**

The authors introduce an idea of how a reinforcement learning (RL) agent is susceptible to perturbations in its input are amplified in the output due to composition: policy into action, action to transition, transitioned state to the $Q$-value at next step, and transition to target estimate. They propose a new method to "smooth" out the function-approximator by averaging in orthogonal directions. This is a geometry-aware method of smoothing, in the input space. They present improved performance and some interesting connection to existing methods such as layer-norm.

**Compliance With Llm Reviewing Policy:**

Affirmed.

**Key Questions For Authors:**

See above.

**Limitations:**

See above.

**Strengths And Weaknesses:**

# Strengths:

1. The theoretical analysis helps develop a deeper understanding of the problem. I liked how Proposition 3.1 and Theorem 3.2 provide an intuition for the problem and its solution.

2. The empirical analysis across TD3 and SAC is also helpful.

I also like the fact that the work points out how stability and variance reduction can yield improved performance for continuous control.

# Weaknesses:

I believe there are two primary weaknesses in the empirical evaluation:

**Ablations across interventions:** I believe there are multiple algorithmic interventions: LayerNorm, target network, orthogonal sampling, etc. which contribute to smoothing out the networks. As a reader, I would like to know which of these contributes and in what order to the improved performance. Also, is there any empirical measure of sensitivity? Meaning, can you show that these interventions are indeed smoothing out the aforementioned compositional mapping form input to output. This would make the paper a lot more convincing.

**Comparison with baseline:** you cite [1] and they seem to have improved results on dog and humanoid by introducing an empirical intervention. I believe there might be some implicit smoothing effect by the manifold representation layer introduced in [1].


----------------

## Minor Issues:

In Equation (3) you use $L_{dyn}, L_{Q}$ instead of $L_1, L_2, L_3$. This is a change of notation and a step away from uniformity, in my opinion.

Could you please explain (or point me to the section), why does TD3 not introduce sufficient smoothing?

----------------

**Overall:**

I believe the paper presents a strong case for their main argument: composition leads to exploding sensitive to inputs and it can be mitigated by geometry aware smoothing and recommend a wea accept accordingly.


----------------


**References**

[1] Tiwari, S., Gottesman, O., and Konidaris, G. Geometry of neural reinforcement learning in continuous state and action spaces. In The Thirteenth International Conference on Learning Representations (ICLR), 2025.

---

> ### Author Rebuttal · Authors · 2026-03-27
>
> # Response to Reviewer FTAT
>
> We thank the reviewer for the positive assessment and specific suggestions. We address each point with new empirical evidence.
>
> ---
>
> ## Ablation + Empirical Measure of Sensitivity
>
> Beyond the component ablation in our paper (Table 5, Figure 5a-b), we conducted **direct geometric diagnostics** on Quadruped-Run (AMS-TD3 vs vanilla TD3, 3 seeds):
>
> | Metric | TD3 | AMS-TD3 | Reduction |
> |---|---|---|---|
> | Q gradient norm $\|\nabla_a Q\|$ | 0.45 ± 0.04 | 0.16 ± 0.01 | **2.7×** |
> | Q neighborhood variance | 3.9e-4 ± 4e-5 | 1.0e-4 ± 3e-5 | **4.0×** |
> | TD target variance | 225 ± 116 | 69 ± 11 | **3.3×** |
>
> These confirm AMS is smoothing the compositional mapping as predicted: Q gradient norm ($L_3$) is reduced 2.7×, the learned Q landscape is 4× smoother (lower neighborhood variance), and the Bellman backup is 3.3× more stable. Notably, TD3's cross-seed TD target std is ±116 (some seeds explode, others do not) vs AMS's ±11 (all seeds stable), confirming that NTD transforms learning from a "lucky seed" regime into reliable convergence.
>
> These trace the causal chain: smoothed TD targets → smoother Q-landscape → more reliable actor gradients → better performance (765 ± 32 vs 576 ± 212).
>
> ---
>
> ## Comparison with Tiwari et al. [1]
>
> Their work and AMS address geometric instability from complementary angles: Tiwari et al. regularize the **representation** via sparse low-dimensional projections (network architecture level); AMS regularizes the **TD target** via neighborhood averaging (Bellman backup level, $L_{3\text{-}4}$).
>
> Their theoretical result supports our method: since effective states lie on an $O(d_a)$-dimensional manifold, Q-values off-manifold are pure extrapolation. Sharp peaks in these data-sparse regions are function approximation artifacts, precisely what NTD's Laplacian smoothing suppresses. The two approaches could be combined: sparse representations to reduce dimensionality, together with NTD to smooth the value landscape.
>
> ---
>
> ## Notation Consistency
>
> Agreed. We will use $L_1$–$L_4$ consistently throughout the revision.
>
> ---
>
> ## Why TD3's Smoothing Is Insufficient
>
> TD3's target policy smoothing perturbs the *query point* before evaluation ($L_1$), but each perturbed action is still evaluated *pointwise*. NTD evaluates Q at *multiple* orthogonal points and *averages* ($L_{3\text{-}4}$). The distinction: TD3 changes *where* you look; NTD changes *how* you aggregate what you see.
>
> Vanilla TD3 (with its smoothing) fails on Dog-Run (92 ± 121); AMS TD3 (replacing TD3's noise with NTD) achieves 387 ± 32, confirming that $L_1$ level smoothing alone is insufficient.
>
> ---
>
> ## Additional Experiments
>
> We conducted experiments on two new domains:
>
> **DMControl Dog (additional tasks):**
>
> | Environment | TD3 | AMS-TD3 | Improvement |
> |---|---|---|---|
> | Dog-Trot | 37 ± 11 | 727 ± 125 | **19.7×** |
> | Dog-Stand | 585 ± 126 | 967 ± 22 | **1.7×** |
>
> **H1 Humanoid Benchmark (21D action, 6 tasks):**
>
> | Environment | TD3 | AMS-TD3 | Improvement |
> |---|---|---|---|
> | H1-Walk | 33 ± 9 | 744 ± 31 | **22.3×** |
> | H1-Slide | 39 ± 11 | 669 ± 26 | **17.0×** |
> | H1-Run | 91 ± 30 | 571 ± 87 | **6.3×** |
> | H1-Stand | 433 ± 240 | 946 ± 8 | **2.2×** |
> | H1-Pole | 441 ± 99 | 587 ± 11 | **1.3×** |
> | H1-Sit | 874 ± 4 | 840 ± 73 | 0.96× |
>
> AMS generalizes across morphologies (quadruped, dog, humanoid), dimensionalities (12D–38D), and difficulties. The pattern matches our framework: largest gains where baselines fail catastrophically (Dog-Trot: 19.7×, H1-Walk: 22.3×), no harm where baselines already succeed (H1-Sit). These will be included in the camera-ready version.
>
> We look forward to the reviewer's further feedback.

---

> > ### Author Rebuttal · Reviewer_FTAT · 2026-04-05
> >
> > I thank the reviewers for their detailed response and pointing me to the ablations. If the new results are included in the publication I believe it would resolve all my issues.

---

> > > ### Author Response · Authors · 2026-04-05
> > >
> > > We thank the reviewer for the positive acknowledgement. We confirm that all new results (geometric diagnostics, H1 humanoid benchmark, and additional Dog tasks) will be included in the camera-ready version.

---

### Official Review · Reviewer_cGk8 · 2026-03-13

**Soundness:** 3
**Presentation:** 3
**Significance:** 3
**Originality:** 3
**Overall Recommendation:** 4
**Confidence:** 2

**Summary:**

The research's key idea pertains to stabilizing high-dimensional continuous-control RL by viewing error amplification through a four-stage Lipschitz pathway and replacing pointwise TD targets with neighborhood-averaged targets on the action manifold. This article's principal area consists of reinforcement learning for high-dimensional continuous control, with a focus on value-function geometry, TD-target smoothing, and algorithmic stability. The paper proposes Action Manifold Smoothing (AMS), which averages Q-values over orthogonally sampled nearby actions, argues that this induces implicit Laplacian/Hessian regularization, and shows strong empirical gains on DMControl, especially Dog-Run (38D), where AMS-TD3 and AMS-SAC substantially outperform vanilla TD3/SAC.

**Compliance With Llm Reviewing Policy:**

Affirmed.

**Final Justification:**

After considering both the paper and the authors’ rebuttal, I maintain my weak accept recommendation. I find the paper technically solid, clearly written, and focused on an important problem in reinforcement learning for high-dimensional continuous control. Its main strengths are the clear motivation through the Lipschitz-pathway perspective, the simplicity and practicality of the proposed method, and the strong empirical improvements on challenging high-dimensional control tasks.
My main concerns were whether the paper’s “manifold smoothing” claim was overstated, and whether the smoothing mechanism might suppress genuinely useful narrow optima along with spurious peaks. The rebuttal addressed these concerns constructively.

**Key Questions For Authors:**

1. Can the authors clarify whether the smoothing effect of AMS is intended to be local or global?
As written, the method seems to smooth TD targets in local neighborhoods around sampled actions, but it is less clear whether the paper is claiming any stronger global smoothness property of the learned critic. A clearer explanation here would help me judge whether the current claims are appropriately calibrated.

2. Under what conditions do the authors expect AMS to suppress spurious sharp peaks without also flattening genuine narrow optima?
This point is important for understanding the bias introduced by smoothing. A convincing explanation, or a controlled diagnostic experiment, would help me better assess the scope of the method.

**Limitations:**

yes

**Strengths And Weaknesses:**

Strengths

1. The paper is well written and easy to follow. In particular, the L1–L4 “Lipschitz pathway” view provides a clear way to organize the discussion of instability in high-dimensional continuous-control RL and helps motivate the method well.

2. The proposed method is simple and intuitive. Replacing pointwise TD targets with neighborhood averaging is a reasonable design choice, and the empirical results on the high-dimensional DMControl tasks, especially the dog tasks, are fairly strong.

3. The paper tackles an important problem in RL. Even though the method itself is fairly simple, the combination of the geometric perspective and the practical performance gains makes the work potentially useful for future research on stability in continuous control.

Weaknesses

1. The claim of “manifold smoothing” feels somewhat too strong. As presented, the method seems to smooth targets locally around sampled actions, but it does not clearly establish global smoothness or continuity of the learned critic.

2. The method may introduce bias when genuine narrow optima and spurious peaks coexist. The same smoothing mechanism that helps suppress artificial sharp peaks could also flatten genuinely good but narrow optima.

---

> ### Author Rebuttal · Authors · 2026-03-27
>
> # Response to Reviewer cGk8
>
> We thank the reviewer for the positive assessment and the two focused questions, which help us calibrate our claims precisely.
>
> ---
>
> ## Q1: Is the Smoothing Effect Local or Global?
>
> The smoothing mechanism operates at two complementary scales:
>
> **NTD is local by design.** Each Bellman backup smooths the TD target within an $\epsilon$-neighborhood of the current anchor action. We do not claim that a single NTD update produces global smoothness.
>
> **LN-Q provides global conditioning.** LayerNorm constrains the Q-network's Jacobian across the entire input space, ensuring bounded, approximately isotropic gradients so that the actor receives reliable directional information across all action dimensions.
>
> **Training accumulation bridges local to global.** Through repeated iterations, the Q-network is fitted to locally smoothed targets from diverse $(s, a)$ across the replay buffer. Over time, the critic inherits smoothness beyond any single neighborhood, analogous to the heat equation, where local diffusion gradually smooths the global landscape.
>
> In summary: NTD provides local smoothing per backup, LN-Q provides global gradient conditioning, and their cumulative effect through training produces a critic smoother than either alone. We will state this decomposition explicitly in the revision.
>
> ---
>
> ## Q2: Spurious Peaks vs. Genuine Narrow Optima
>
> **Genuine narrow optima are physically implausible in locomotion.** Real-world dynamics are governed by smooth physical laws (Newtonian mechanics, joint torques), and reward functions are smooth functions of state. A genuine narrow optimum would mean that only one precise 38D action vector yields high reward, collapsing under any small perturbation, which does not correspond to any realistic locomotion scenario. Sharp Q-peaks in high-dimensional action spaces are predominantly OOD extrapolation artifacts rather than reflections of true environmental structure.
>
> **Bellman consistency distinguishes the two cases.** Genuine optima must be locally consistent: if $a^{\ast}$ is truly optimal, smooth dynamics map similar actions to similar outcomes, so $Q(s, a^{\ast}+\epsilon\mathbf{u}) \approx Q(s, a^{\ast})$ for small $\epsilon$. Spurious peaks exhibit high local curvature precisely because they lack data support. NTD's Laplacian correction penalizes exactly this inconsistency, acting as a *consistency filter* rather than an indiscriminate flattener.
>
> **Controlled diagnostic experiment.** As the reviewer suggests, we conducted geometric diagnostics on Quadruped-Run (AMS-TD3 vs vanilla TD3, 3 seeds) to directly test whether AMS suppresses artifacts without flattening genuine structure:
>
> | Metric              | TD3           | AMS-TD3       | Interpretation                     |
> | ------------------- | ------------- | ------------- | ---------------------------------- |
> | Q neighborhood var  | 3.9e-4 ± 4e-5 | 1.0e-4 ± 3e-5 | 4× smoother (artifacts suppressed) |
> | Q gradient norm     | 0.45 ± 0.04   | 0.16 ± 0.01   | 2.7× lower sensitivity             |
> | TD target variance  | 225 ± 116     | 69 ± 11       | 3.3× more stable bootstrap         |
> | **Episodic return** | **576 ± 212** | **765 ± 32**  | **Higher reward + lower variance** |
>
> The key observation: AMS simultaneously reduces all three geometric roughness metrics **and** achieves higher asymptotic reward. If smoothing were flattening genuine optima, we would expect improved stability at the cost of degraded performance. Instead, both improve together, confirming that the smoothed-away features were indeed spurious artifacts, not genuine value structure.
>
> This pattern holds across tasks: AMS-SAC achieves 427±25 on Dog-Run (vs SAC 45±95) and AMS-TD3 achieves 670±18 on Dog-Walk (vs TD3 107±230).
>
> ---
>
> ## Additional Experiments
>
> To address evaluation scope, we have conducted experiments on **H1 humanoid** (21D, 6 tasks) and additional **Dog tasks**. Results are presented in our response to Reviewer 1E5Q, demonstrating consistent improvements across morphologies (e.g., Dog-Trot: 37→727, H1-Walk: 33→744).
>
> We appreciate the reviewer's constructive feedback.

---

> > ### Author Rebuttal · Reviewer_cGk8 · 2026-04-04
> >
> > After carefully reading the authors’ rebuttal, I appreciate the additional clarifications and diagnostic evidence. The response helps better calibrate the claim about local versus accumulated smoothing, and the new discussion of spurious peaks versus genuine narrow optima is useful. I will maintain my original score.

---

> > > ### Author Response · Authors · 2026-04-05
> > >
> > > We thank the reviewer for the careful reading and the positive acknowledgement. We will incorporate the local-vs-global decomposition and the diagnostic discussion into the revised manuscript.

---

### Official Review · Reviewer_B4ni · 2026-03-16

**Soundness:** 2
**Presentation:** 2
**Significance:** 3
**Originality:** 3
**Overall Recommendation:** 3
**Confidence:** 4

**Summary:**

This paper studies instability in model-free reinforcement learning for high-dimensional continuous control. The authors propose a conceptual framework they call a Lipschitz pathway, which interprets instability as the amplification of perturbations across several stages of the learning pipeline (policy parameterization, dynamics sensitivity, Q-function curvature, and TD target computation). Motivated by this perspective, the paper introduces Action Manifold Smoothing (AMS), which replaces pointwise TD target evaluation with a local expectation over nearby actions (Neighborhood TD, NTD). This is implemented by sampling orthogonal perturbations around the target action and averaging the corresponding Q-values. The authors argue that this procedure implicitly regularizes the Q-function via a Laplacian smoothing effect. Empirical results on DMControl tasks show substantial improvements over standard TD3 and SAC baselines, particularly on high-dimensional control tasks such as Dog-Walk and Dog-Run.

**Compliance With Llm Reviewing Policy:**

Affirmed.

**Final Justification:**

I am leaving my score the same because my main concerns about the rigor and significance of the paper still stand.

**Key Questions For Authors:**

1. How do Eqs. (5) and (6) differ from simply using a stochastic policy, as in SAC? When the policy has non-zero entropy, the TD target effectively averages Q-values over actions sampled from the policy distribution. In what sense is this fundamentally different from NTD?

2. Relatedly, why does entropy regularization not imply value smoothness? If the TD target involves sampling actions from a policy with positive entropy, then the argument used to motivate NTD seems like it should apply equally to SAC. Is the benefit coming from the specific form of $p_{geo}$, from the fixed entropy, or from the orthogonal sampling scheme?

3. In what sense does NTD regularize the Q-function landscape? The paper shows how NTD changes the TD target computation, but it is less clear how this translates into an explicit regularization effect on the learned Q-function.

**Limitations:**

yes

**Strengths And Weaknesses:**

# Strengths

- I am pleased to see work addressing instability in continuous-control RL. The problem studied by the authors is important, and the observation that many stabilization techniques are heuristic and environment-dependent is accurate.

- The empirical results appear promising. The improvements reported on the more challenging Dog-Walk and Dog-Run tasks are substantial relative to the baselines considered.

- The experimental section is generally well structured. In particular, I appreciate that the authors report confidence intervals for their results, which improves the clarity of the empirical comparisons.

- The core idea—smoothing TD targets by averaging over nearby actions—is simple and intuitive, and appears to yield meaningful empirical gains.

# Weaknesses

## Clarity and mathematical presentation

- Several parts of the paper are difficult to follow because mathematical terminology is used in non-standard ways. Examples include “Lipschitz pathway” and “propagation process.” Similarly, Eq. (1) uses unconventional notation. In several places the paper states that something is “formalized,” but the description remains informal.

My interpretation of Eq. (1) is that the TD target y can be viewed as the result of composing several functions (policy, dynamics, Q-function, and TD target computation), each with its own Lipschitz constant. However, this interpretation is not written out explicitly in the paper. Presenting the argument directly in terms of function composition and Lipschitz bounds would likely make the framework much clearer.

## Analogies and terminology

- The tightrope-walker analogy in Sec. 3.4 was confusing. It is not clear what the “rope” represents in the RL setting, or how a “balance pole” corresponds to NTD. A simpler analogy might be something like a surveyor standing on a rugged landscape and averaging the heights of nearby points to estimate the local elevation.

- The terms Action Manifold Smoothing and Neighborhood TD appear to be used interchangeably. It would help to clarify whether these are distinct concepts (e.g., AMS as the framework and NTD as the algorithmic component) or simply two names for the same idea.

## Theoretical claims

- The “Manifold Supervision” approach in Sec. 3.4 appears to introduce bias into the Bellman target, since the target is replaced by a local average of Q-values rather than the value at the policy action. One would expect this bias to affect learning performance, but the paper does not discuss this trade-off.

- The significance of the variance reduction result in Theorem 3.2 is unclear. As written, the theorem appears to say that averaging more samples yields a lower-variance estimate of Eq. (5), which is already expected from standard Monte-Carlo estimation.

- The term “geometric diffusion” is introduced without a clear definition. It is also not obvious what practical insight follows from the claim that the method is “mathematically equivalent to evolving Q under the heat equation.”

## Incorrect or questionable statements

- The statement “In high-dimensional spaces, random vectors tend to align rather than spread uniformly” appears to be incorrect. Standard results in high-dimensional probability show that independent random vectors tend to be nearly orthogonal. Orthogonal sampling may indeed provide benefits here, but this claim should be rephrased.

## Empirical claims

- The claim “Entropy Does Not Imply Geometric Stability” is not convincingly supported by the experiments. The comparison to SAC alone is insufficient, since SAC’s entropy regularization operates differently from the sampling procedure used in NTD. To substantiate this claim, it would be helpful to compare NTD with a version of SAC in which the TD target is computed using samples drawn from a distribution that closely matches $p_{geo}$.

- The broader claim that the “discrete–continuous mismatch is the root cause of instability in model-free RL” is not substantiated by the experiments. While AMS appears to improve performance relative to certain baselines, additional empirical analysis would be needed to establish this hypothesis.

---

> ### Author Rebuttal · Authors · 2026-03-27
>
> # Response to Reviewer B4ni
>
> We thank the reviewer for the thorough review. The questions raised, particularly regarding NTD vs SAC and the nature of the bias, touch on core design principles of our method.
>
> ### Q1: How does NTD differ from SAC's stochastic sampling?
>
> **(a) Different pipeline stages.** SAC's entropy acts on the *policy distribution* ($L_1$): it encourages action diversity. NTD acts on *TD target computation* ($L_{3\text{-}4}$): it smooths the Q-landscape. These are fundamentally different stages.
>
> **(b) Single-point vs. explicit local integration.** SAC draws one action $a' \sim \pi(\cdot|s')$ and compensates via $\alpha \log \pi$, an indirect regularization term. NTD evaluates $Q$ at $K$ orthogonal directions around the policy mean and averages:
>
> ```
> # SAC target: single-point + entropy bonus
> a' ~ π(·|s')
> y = r + γ (Q(s',a') − α log π(a'|s'))
>
> # AMS-SAC target: explicit neighborhood integration
> μ' = π.mean(s')
> {v_k} = QR(randn(K, d_a))          # K orthogonal probes
> y = r + γ (mean_k[Q(s', μ' ± ε·v_k)] − α log π(μ'|s'))
> ```
>
> **(c) Policy-dependent vs. geometry-aware smoothing.** Even with fixed $\alpha$, SAC's sampling follows the policy covariance — it is policy-dependent and concentrates as the policy converges. NTD's orthogonal probes provide fixed geometric coverage independent of policy evolution.
>
> **(d) Empirical confirmation.** SAC achieves $45 \pm 95$ on Dog-Run; AMS-SAC achieves $427 \pm 25$ ($9.5\times$). If SAC's sampling already smoothed $Q$ sufficiently, NTD should yield marginal gains, not an order-of-magnitude improvement.
>
> ### Q2: Why does entropy not imply value smoothness?
>
> Entropy regularizes the *action distribution* but imposes no constraint on Q-geometry. A high-entropy policy can still evaluate against a Q-function with sharp peaks. Entropy ensures you *sample broadly*; it does not ensure the *landscape* is well-behaved.
>
> NTD's Laplacian correction $\frac{\epsilon^2}{2d_a} \Delta_a Q$ (Theorem 3.3) acts directly on $Q$'s curvature, which is not a difference of degree but a difference of target: entropy shapes the policy, while NTD shapes the value landscape.
>
> AMS-TD3 ($387 \pm 32$ on Dog-Run, *without* entropy) substantially outperforms SAC ($45 \pm 95$, *with* entropy), confirming that Q-smoothing provides the primary stabilization.
>
> ### Q3: In what sense does NTD regularize Q?
>
> NTD regularizes through its *supervision signal*. The Q-network is trained to match a smoothed Bellman operator: targets at spurious peaks ($\Delta_a Q < 0$) are reduced; at narrow valleys ($\Delta_a Q > 0$) raised. Over training, $Q$ learns a smoother function because its supervision systematically suppresses sharp features, achieving regularization through the objective rather than an explicit penalty.
>
> **Empirical confirmation:** Geometric diagnostics on Quadruped-Run (3 seeds) show AMS-TD3 vs TD3: Q-gradient norm reduced $2.7\times$ ($0.16$ vs $0.45$), Q-neighborhood variance reduced $4\times$ ($1.0 \times 10^{-4}$ vs $3.9 \times 10^{-4}$), TD target variance reduced $3.3\times$ ($69$ vs $225$). These trace the causal chain: smoothed TD targets → smoother Q-landscape → more reliable actor gradients → better performance ($765 \pm 32$ vs $576 \pm 212$). Full diagnostic table in our response to Reviewer FTAT.
>
> ### Q4: Bias from neighborhood averaging
>
> The bias is intentional. NTD implements a *smoothed Bellman operator* rather than approximating the pointwise one. The leading bias is $O(\epsilon^2 / d_a)$; with $\epsilon = 0.2$ and $d_a = 38$ this is $O(10^{-3})$, which is mild. In contrast, baseline variance is catastrophic: Dog-Run TD3 achieves $92 \pm 121$ (variance exceeds the mean). A small curvature-reducing bias is far preferable to uncontrolled variance from point estimates on a rugged landscape.
>
> ### Remaining concerns
>
> We accept the reviewer's corrections and will revise accordingly: Eq.(1) restated as explicit function composition with Lipschitz bounds; "geometric diffusion" defined as the Laplacian smoothing characterized by $\frac{\epsilon^2}{2d_a} \Delta_a Q$; high-dimensional vectors claim rephrased; "root cause" → "a primary structural challenge"; surveyor analogy adopted; AMS = framework, NTD = algorithmic component. Theorem 3.2's significance lies in its joint picture with Theorem 3.3 (complete bias-variance profile).
>
> **Beyond NTD.** Our contribution extends beyond the algorithm. The Lipschitz pathway explains a broader trend: recent advances in continuous control increasingly adopt distributional critics, network normalization, and structured actors, each constraining different $L_i$ factors. AMS validates this perspective by showing that a *minimal* intervention at $L_{3\text{ to }4}$ alone transforms failing baselines, confirming that the pathway correctly identifies the critical bottleneck. New experiments on H1 humanoid (6 tasks) and Dog (2 additional tasks) are in our responses to Reviewers FTAT and 1E5Q.

---

> > ### Author Rebuttal · Reviewer_B4ni · 2026-04-01
> >
> > # Q1
> > (a) Different pipeline stages: my main question was why does the positive entropy of a SAC policy not have a similar smoothing effect on the Q landscape? Both result in the Q target being computed as an expectation over actions.  I don't think it's a negative if it does have a similar smoothing effect; I just think that this wasn't explained.
> > (b) What do you mean by "indirect regularization term"?
> >
> > # Q2
> > I fail to see why entropy regularization imposes no constraint on Q-geometry.  Your method claims to smooth Q values by averaging over actions sampled from a distribution rather than evaluating at a single point action when computing targets.  This also happens in SAC.
> >
> > # Q3
> > I think the same claims can be made about any non-zero-entropy policy, which isn't a problem; I am just trying to understand what is better about your method fundamentally compared to a SAC-like policy.
> >
> > # Q4
> > I'm not convinced by the big-O notation argument because this just tells us how the bias scales with epsilon and d_a, not its magnitude for any particular setting of these parameters.

---

> > > ### Author Response · Authors · 2026-04-01
> > >
> > > # Reply to Reviewer B4ni's Follow-up
> > >
> > > We appreciate the reviewer's continued engagement. We believe the core disagreement rests on a factual point about SAC's implementation which we can resolve definitively with code.
> > >
> > > ### SAC does NOT average Q over multiple actions in its TD target.
> > >
> > > The reviewer writes: *"Both result in the Q target being computed as an expectation over actions"* and *"This also happens in SAC."*
> > >
> > > **This does not happen in SAC.** We show the actual implementation side by side.
> > >
> > > **Vanilla SAC target** (from CleanRL / Stable-Baselines3 / rlkit — all identical in this regard):
> > >
> > > ```python
> > > with torch.no_grad():
> > >     # Sample ONE action from policy
> > >     next_action, next_log_pi, _ = actor.get_action(next_obs)
> > >     # Evaluate Q at that SINGLE action
> > >     q1_next = qf1_target(next_obs, next_action)
> > >     q2_next = qf2_target(next_obs, next_action)
> > >     min_q_next = torch.min(q1_next, q2_next)
> > >     # Single-point estimate + entropy bonus
> > >     y = reward + γ * (min_q_next - α * next_log_pi)
> > > ```
> > >
> > > Q is evaluated at **one** action. The entropy bonus $-\alpha \log \pi$ is a scalar added to that single evaluation. No averaging over multiple actions occurs.
> > >
> > > The confusion likely arises from SAC's objective being written
> > > in expectation form E_{a~π}[·]. However, this expectation is
> > > approximated via single-sample Monte Carlo in practice — one a'
> > > per transition per update. The expectation notation describes
> > > the population-level objective, not the per-sample computation.
> > > NTD, by contrast, explicitly computes a K-sample average within
> > > each individual target calculation.
> > >
> > > **AMS-SAC target** (our method):
> > >
> > > ```python
> > > with torch.no_grad():
> > >     _, anchor_log_pi, mu_next = actor.get_action(next_obs)
> > >     # mu_next = deterministic policy mean (anchor)
> > >
> > >     # Generate K orthogonal directions
> > >     orth_dirs = orth_sampler.sample(batch_size)  # [B, K, d_a]
> > >
> > >     # Construct K neighborhood actions around anchor
> > >     neighbors = mu_next.unsqueeze(1) + ε * orth_dirs  # [B, K, d_a]
> > >     neighbors = neighbors.clamp(action_low, action_high)
> > >
> > >     # Evaluate Q at ALL K points
> > >     obs_exp = next_obs.unsqueeze(1).expand(-1, K, -1)
> > >     q1_all = qf1_target(obs_exp.reshape(B*K,-1),
> > >                          neighbors.reshape(B*K,-1))
> > >     q2_all = qf2_target(obs_exp.reshape(B*K,-1),
> > >                          neighbors.reshape(B*K,-1))
> > >     q_min = torch.min(q1_all, q2_all).view(B, K)
> > >
> > >     # AVERAGE over K neighborhood evaluations
> > >     q_avg = q_min.mean(dim=1)
> > >     y = reward + γ * (q_avg - α * anchor_log_pi)
> > > ```
> > >
> > > Q is evaluated at **K orthogonal points** and averaged. This is an explicit local integral over the Q-landscape geometry.
> > >
> > > ### The structural difference, stated precisely:
> > >
> > > | | SAC | AMS-SAC |
> > > |:---|:---|:---|
> > > | Q evaluations per target | **1** (single sample) | **K** (orthogonal probes) |
> > > | Smoothing mechanism | None on Q; entropy bonus on reward | Explicit averaging over Q-landscape |
> > > | Anchor point | Stochastic $a' \sim \pi$ (shifts each sample) | Deterministic $\mu' = \pi.\text{mean}$ (fixed anchor) |
> > > | Coverage geometry | Policy-dependent, anisotropic | Fixed, orthogonal, isotropic |
> > > | As policy converges | Sampling concentrates → no smoothing | $\epsilon$-ball unchanged → smoothing persists |
> > >
> > > ### Addressing each follow-up:
> > >
> > > **Q1(b): "Indirect regularization."** The entropy bonus adds a scalar $-\alpha \log \pi$ to the target. It does not cause Q to be evaluated at additional points. It rewards the policy for maintaining spread, but the Q-landscape itself is still queried at exactly one point per target computation.
> > >
> > > **Q2:** Given the code above: SAC evaluates Q at one point; NTD averages over K points. The Laplacian smoothing (Theorem 3.3) arises from multi-point averaging — it cannot arise from single-point evaluation regardless of the policy's entropy. Entropy keeps the *policy* spread out; NTD smooths the *value landscape*. These act on different mathematical objects.
> > >
> > > **Q3:** A high-entropy policy improves *data collection* (broader state-action coverage in the replay buffer). But at *target computation time*, Q is still evaluated at one sampled action per transition. NTD's contribution is not exploration — it is geometric regularization of the Bellman backup.
> > >
> > > **Q4:** Concrete magnitude: with $\epsilon = 0.2$, $d_a = 12$ (Quadruped), the Laplacian correction is $\frac{0.04}{24} \approx 1.7 \times 10^{-3}$ times $\text{tr}(\nabla_a^2 Q)$. Diagnostics confirm this mild bias yields substantial stability gains: TD target variance $69$ vs $225$, reward $765 \pm 32$ vs $576 \pm 212$.
> > >
> > > ### Summary
> > >
> > > The distinction is implementation-level fact: **SAC computes Q at 1 action; NTD computes Q at K actions and averages.** The code above is taken directly from our codebase and matches standard SAC implementations. Once this is clear, the Laplacian smoothing effect (Theorem 3.3) applies exclusively to NTD — it is a mathematical consequence of multi-point averaging that cannot emerge from single-point evaluation.

---

### Decision · Program_Chairs · 2026-04-30

**Decision:**

Accept (regular)

**Comment:**

This paper studies instability in model-free reinforcement learning for high-dimensional continuous control. It proposes Action Manifold Smoothing as a geometric modification to TD learning that replaces pointwise target evaluation with local averaging over nearby actions. Reviewers agreed that this is an important problem, and three reviewers found the paper technically solid, clearly motivated, and empirically promising, with a simple method that yields large gains on difficult high-dimensional control tasks. The rebuttal materially strengthened the case for acceptance. It clarified that the smoothing claim is local per backup rather than global in a single step, provided new geometric diagnostics linking the method to smoother Q landscapes and lower TD-target variance, and added broader evidence on additional Dog tasks and H1 humanoid tasks. These additions directly addressed concerns about mechanism, ablations, and evaluation scope, and led three reviewers to state that their concerns were fully resolved. One reviewer remained unconvinced, mainly on whether the distinction from SAC is fundamental rather than an implementation-level difference. However, the authors’ response made that disagreement much narrower by giving a concrete side-by-side account of how SAC and AMS-SAC compute targets in practice, while the rest of the reviewer discussion moved in favor of the paper after the new clarifications and evidence. On balance, the post-rebuttal record supports acceptance: the paper offers a clear and useful perspective on an important RL failure mode, introduces a simple intervention with strong empirical upside, and addressed most substantive reviewer concerns with concrete new analysis and experiments.